# PA3FF: P̲art-A̲ware Dense 3̲D F̲eature F̲ield For Generalizable Articulated Object Manipulation

**Yue Chen**[1*]   **Muqing Jiang**[1*]   **Kaifeng Zheng**[2*]
**Jiaqi Liang**[1]   **Chenrui Tie**[3]   **Haoran Lu**[1]   **Ruihai Wu**[1†]   **Hao Dong**[1†]
[1]Peking University   [2]Beijing Institute of Technology   [3]National University of Singapore

## Abstract

Articulated object manipulation is essential for various real-world robotic tasks, yet generalizing across diverse objects remains a major challenge. A key to generalization lies in understanding functional parts (e.g., door handles and knobs), which indicate where and how to manipulate across diverse object categories and shapes. Previous works attempted to achieve generalization by introducing foundation features, while these features are mostly 2D-based and do not specifically consider functional parts. When lifting these 2D features to geometry-profound 3D space, challenges arise, such as long runtimes, multi-view inconsistencies, and low spatial resolution with insufficient geometric information. To address these issues, we propose **Part-Aware 3D Feature Field (PA3FF)**, a novel dense 3D feature with part awareness for generalizable articulated object manipulation. PA3FF is trained by 3D part proposals from a large-scale labeled dataset, via a contrastive learning formulation. Given point clouds as input, PA3FF predicts a continuous 3D feature field in a feedforward manner, where the distance between point features reflects the proximity of functional parts: points with similar features are more likely to belong to the same part. Building on this feature, we introduce the **Part-Aware Diffusion Policy (PADP)**, an imitation learning framework aimed at enhancing sample efficiency and generalization for robotic manipulation. We evaluate PADP on several simulated and real-world tasks, demonstrating that PA3FF consistently outperforms a range of 2D and 3D representations in manipulation scenarios, including CLIP, DINOv2, and Grounded-SAM, achieving state-of-the-art performance. Beyond imitation learning, PA3FF enables diverse downstream methods, including correspondence learning and segmentation tasks, making it a versatile foundation for robotic manipulation. Project page: https://pa3ff.github.io/.

## 1 Introduction

The next generation of assistive robots must possess the generalization ability to manipulate objects across a broad range of scenarios with ease and adaptability. To achieve this goal , recent studies (Black et al., 2024; Intelligence et al., 2025; Team et al., 2024; Kim et al., 2024b) leverage the power of 2D vision-language foundation models (e.g., CLIP (Radford et al., 2021), DINOv2 (Oquab et al., 2023), SigLIP (Zhai et al., 2023)) to improve performance and generalization in robotic manipulation policies. However, these representations inherently lack 3D geometry and spatial continuity, which are crucial for reasoning about object shapes, part configurations, and affordance in manipulation tasks (Zhu et al., 2024; Zhang et al., 2023; Ke et al., 2024a).

Some recent works attempt to lift 2D features into 3D feature fields via multi-view fusion or neural rendering (Kerr et al., 2023a; Shen et al., 2023; Lin et al., 2023; Rashid et al., 2023; Ze et al., 2023). While these methods improve the understanding of 3D objects and scenes, they are not native 3D representations, and thus usually suffer from problems such as **long inference times** (even minutes),

---

*Equal contribution
†Corresponding author

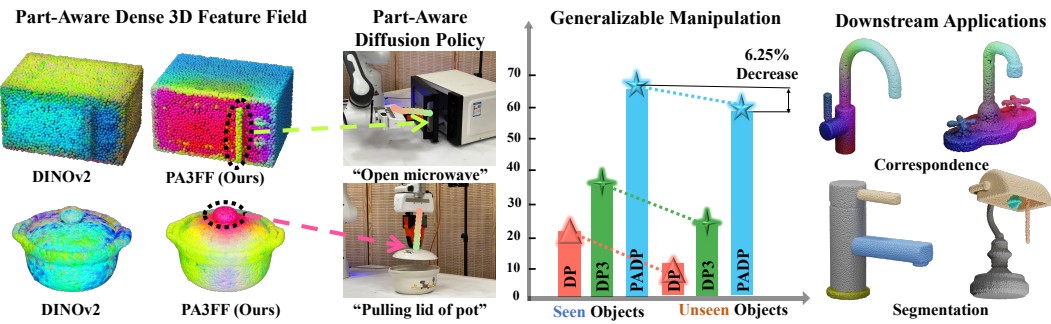

Figure 1: (1) We propose PA3FF, a feedforward model that predicts part-aware 3D feature fields for 3d shapes. (2) We propose a part-aware diffusion policy, which leverages PA3FF, that can efficiently generalize to unseen objects. (3) PA3FF exhibits consistency across shapes, enabling various downstream applications such as correspondence and segmentation.

**feature inconsistency across views** (Wang et al., 2024b), and **limited spatial resolution** (Fu et al., 2024), making them impractical for dense, fine-grained and real-time robotic manipulation.

In this paper, we propose **Part-Aware 3D Feature Field (PA3FF)**, a novel 3D-native representation designed to encode dense, semantic, and part-aware features directly from point clouds. PA3FF predicts a continuous 3D feature field in a feedforward manner, where the distance between features reflects the part-level awareness—points with similar features are more likely to belong to the same part. We leverage Sonata, a model pretrained on 140k point clouds with self distillation, to provide rich 3D geometric priors for our proposed representation. To enhance the part-awareness, we incorporate a contrastive learning framework that establishes consistent relationships between 3D part-level features and their corresponding semantic counterparts by a collection of public dataset (PartNet-Mobility (Xiang et al., 2020; Mo et al., 2018), 3DCoMPaT (Slim et al., 2025), PartObjaverse-Tiny (Yang et al., 2024b)).

To demonstrate the power of PA3FF in robotic manipulation scenarios, we introduce **Part-Aware Diffusion Policy (PADP)**, a 3D point cloud-based visuomotor policy that integrates PA3FF with a diffusion policy architecture, as shown in Figure 1 (Left). The part awareness and generalization capability to novel shapes empower the highly sample-efficient and generalizable manipulation behaviors across objects. As shown in Figure 1 (Right), beyond imitation learning, PA3FF enables diverse downstream methods, including correspondence learning, along with key-point proposals for planning constraints, making it a versatile foundation for robotic manipulation.

We evaluate PA3FF and PADP on a broad spectrum of robotic manipulation tasks from the PartInstruct (Yin et al., 2025), as well as in the real world. PADP sets a new state-of-the-art on PartInstruct with a 9.4% absolute gain, outperforming existing 2D and 3D representations with Diffusion Policy (Chi et al., 2023), such as CLIP (Radford et al., 2021), DINOv2 (Oquab et al., 2023) and Grounded-SAM (Ren et al., 2024). We further show PADP significantly surpasses a strong baseline (GenDP (Wang et al., 2024c)) from 8 real-world tasks, offering 18.75% increment. We also provide a detailed analysis of how well and why our method can generalize to novel instances.

In summary, our contributions include:

- We introduce **PA3FF**, a 3D-native representation that encodes dense, semantic, and functional part-aware features directly from point clouds.

- We develop **PADP**, a diffusion policy that leverages PA3FF for generalizable manipulation with strong sample efficiency.

- PA3FF can further enable diverse downstream methods, including correspondence learning and segmentation, making it a versatile foundation for robotic manipulation.

- We validate our approach on 16 PartInstruct and 8 real-world tasks, where it significantly outperforms prior 2D and 3D representations (CLIP, DINOv2, and Grounded-SAM), offering a 15% and 16.5% increase.

## 2 RELATED WORK

**3D Semantic Representation in Robotic Manipulation.** One common approach to 3D semantic representation involves extracting functional or affordance information from observations Paulius et al. (2016); Kokic et al. (2017); Wu & Zhao (2022); Zhao et al. (2023b); Wang et al. (2022); Wu et al. (2023); Wen et al. (2022); Di Palo & Johns (2024); Chen et al. (2024); Wu et al. (2025a); Liang et al. (2026). This line of research focuses on tasks that can be accomplished using motion primitives such as grasping, picking, and placing. However, our diffusion-based policy offers greater flexibility in terms of action representation and task execution. Another approach lifts 2D foundational features (e.g., CLIP, DINOv2) to 3D representations via multi-view fusion (Wang et al., 2024b; Kerr et al., 2023a; Wang et al., 2024c; Ke et al., 2024a; Yang et al., 2024b; Kerr et al., 2023b; Qiu et al., 2024; Zou et al., 2025). However, this method has several notable limitations. First, multi-view fusion in previous approaches is computationally expensive and suffers from inconsistent features across views. Second, these methods often sacrifice spatial resolution for semantic quality in 2D features. For instance, ViT-family models process image tokens as patches, resulting in significantly lower-resolution feature maps (e.g., 14x smaller in DINOv2), which leads to a substantial loss of spatial information. In contrast, our approach leverages visual representations that are pre-trained on large-scale point clouds, which enables our robot to generalize to unseen configurations efficiently.

**Imitation Learning.** Imitation learning (IL) has proven effective in enabling end-to-end robotic training from expert demonstrations via supervised learning (Peng et al., 2020; Radosavovic et al., 2021; Zhao et al., 2023a; Tie et al., 2025; Chi et al., 2023). However, many recent IL methods rely on large-scale datasets to learn robust manipulation policies (Radosavovic et al., 2021; Peng et al., 2020). To improve data efficiency, several works (Tie et al., 2025; Yang et al., 2024a; Wang et al., 2024a) incorporate equivariance into policy architectures, thereby enhancing spatial generalization. Other approaches have explored multi-modal fusion of 3D vision, language instructions, and proprioception (Gervet et al., 2023; Shridhar et al., 2023; Zhang et al., 2023; 2024; Ke et al., 2024a; Wang et al., 2025b). Despite their success, these methods typically predict discrete keyframes rather than continuous control trajectories (e.g., PerAct (Shridhar et al., 2023), Act3D (Gervet et al., 2023), Chained Diffuser (Xian et al., 2023), and 3D Diffuser Actor (Ke et al., 2024a)), limiting their effectiveness in long-horizon or fine-grained manipulation tasks. A method most closely related to ours is GenDP (Wang et al., 2024c), which computes dense semantic fields by measuring cosine similarity between 2D image features and scene observations. This enables category-level generalization, but suffers from two key limitations: (1) its reliance on 2D features from DINOv2 introduces inconsistencies across views; and (2) its semantic representations lack the granularity needed to identify functionally relevant object parts, which are critical for manipulation. In contrast, our method introduces a 3D-native fine-grained feature field that is function-aware, allowing for more accurate localization of interactive parts. Based on this representation, we develop a manipulation policy that not only requires fewer demonstrations but also generalizes across unseen object categories, addressing both data efficiency and generalization in robotic manipulation.

## 3 METHOD

In this section, we cover the different components of our approach, as shown in 2. Initially, we present an overview of Part-Aware 3D Feature Field (PA3FF), and data strategy, training process, along with model architecture in section 3.1. Subsequently, we explore how to leverage this 3D feature field to achieve generalizable articulated object manipulation learning in section 3.2.

### 3.1 PART-AWARE 3D FEATURE FIELD

**Problem Formulation.** Our objective is to design a 3D and part-awareness feature for generalizable articulated object manipulation. Given an input point cloud $\mathcal{P} = \{\mathbf{p}_i \in \mathbb{R}^3\}_{i=1}^N$, this model predicts a continuous 3D feature field $f : \mathbb{R}^3 \to \mathbb{R}^n$ that encodes the part structure and their hierarchy in a feedforward manner. This feature field assigns each point $\mathbf{p} \in \mathcal{P}$ an $n$-dimensional latent feature vector $f(\mathbf{p})$, resulting in a per-point embedding of the input point clouds. The notion of parts is captured by the proximity of features in this latent space: points $\mathbf{p}_a$ and $\mathbf{p}_b$ that belong to the same part should have similar features, i.e., $f(\mathbf{p}_a) \approx f(\mathbf{p}_b)$.

**Backbone for 3D Feature Extraction.** In this stage, we aim to get a 3D feature extraction backbone that leverages the geometric cues of 3D objects and learns 3D priors from a large-scale 3D dataset.

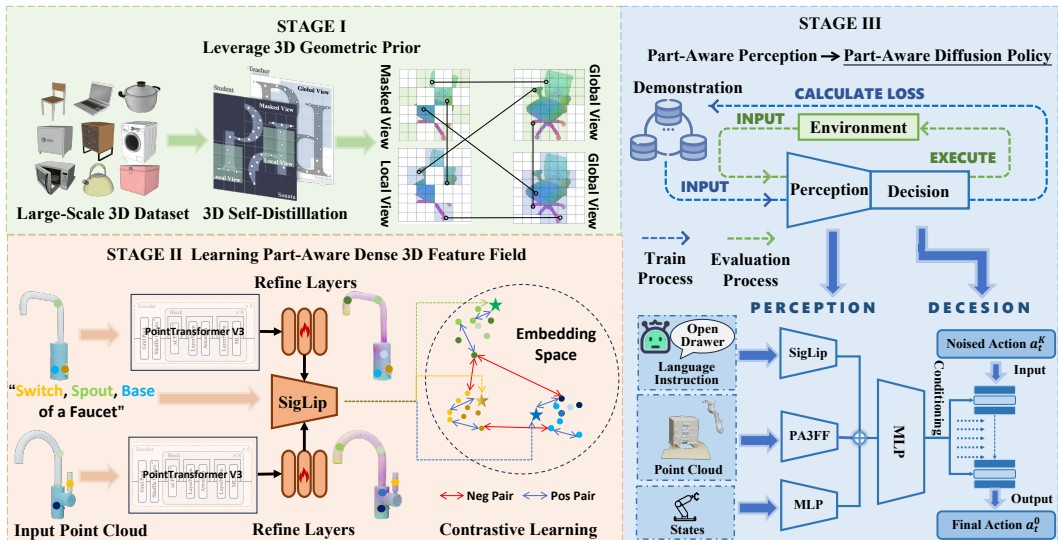

Figure 2: Overview of our Learning Framework. (1) *Pretraining the PTv3 backbone* to extract part-aware 3D features. (2) *Feature refinement via contrastive learning* across objects to enhance part-level consistency and distinctiveness. (3) *Downstream usage* by integrating the refined features into a diffusion policy for action generation.

Unlike prior work He et al. (2024); Yang et al. (2024b); Kim et al. (2024a); Liu et al. (2023); Zhou et al. (2023) that relies on per-shape optimization to lift or distill 2D predictions or priors, we instead leverage Sonata (Wu et al., 2025b), a self-supervised pre-trained Point Transformer V3 (Wu et al., 2024). We then employ Sonata with its pre-trained weights as our feature extractor $f(\mathbf{p})$ to extract multi-scale features from point clouds. The key advantage of our approach lies in addressing limitations present in prior methods that rely on 2D feature distillation. First, multi-view fusion in previous approaches is computationally slow and suffers from the problem of inconsistent features across views. Second, these methods typically sacrifice spatial resolution for semantic quality in 2D features—for example, ViT-family models process image tokens as patches, resulting in much lower-resolution feature maps (e.g., 14x smaller in DINOv2), leading to a significant loss of spatial information. In contrast, Sonata allows for 3D dense feature extraction that maintains both geometric accuracy and semantic information. This approach offers several advantages: (a) efficient, feedforward inference; (b) consistent and complete 3D feature fields that generalize well across objects; and (c) per-point dense features that capture accurate geometric cues. However, Sonata was originally trained on scene-level data and is not directly tailored for object- or part-level representation learning. Specifically, its backbone, PointTransformer v3 (PTv3), is designed for large-scale scenes, using aggressive downsampling to expand receptive fields and reduce computational cost. In contrast, object-level inputs are smaller in both point count and spatial extent, making such downsampling suboptimal. To adapt Sonata for our task, we remove most downsampling layers in PTv3 and instead deepen the network by stacking additional transformer blocks, which enhances detail preservation and improves feature abstraction. Importantly, our overall framework is model-agnostic and can accommodate more advanced 3D feature extraction. More details can be found in Appendix A

**Learning Part-Aware Dense 3D Feature Field.** Building on the promising 3D priors we have obtained, we aim to enhance these representations by incorporating part-aware semantic features. To achieve this, we introduce a contrastive learning framework that effectively establishes consistent relationships between 3D part-level features and their corresponding semantic counterparts.

We design two complementary loss functions to achieve this goal. The first one is **geometric loss**, which focuses on the spatial relationships between points within the same part and between different parts. It encourages the model to bring points from the same part closer together in the feature space while pushing points from different parts apart. Given a set of $N$ feature/label pairs $\{\boldsymbol{f}_k, a_k\}_{k=1...N}$, and assumes there are $N_{a_k}$ samples sharing label $a_k$. The geometric loss is derived from the Supervised Contrastive Loss (Khosla et al., 2021), a widely-used method in contrastive learning. The geometric loss is defined as:

$$\mathcal{L}_{Geo} = \sum_{i=1}^{N} \frac{-1}{N_{a_i} - 1} \sum_{j=1}^{N} \mathbf{1}_{i \neq j} \cdot \mathbf{1}_{a_i = a_j} \cdot \log \frac{\exp\left(\boldsymbol{f}_i \cdot \boldsymbol{f}_j / \tau\right)}{\sum_{k=1}^{N} \mathbf{1}_{i \neq k} \cdot \exp\left(\boldsymbol{f}_i \cdot \boldsymbol{f}_k / \tau\right)} \tag{1}$$

where $\tau \in \mathbb{R}^+$ denotes the balancing coefficient in SupCon.

In addition to geometric alignment, we introduce a **semantic loss**, which aligns point-level features with semantic representations derived from part names. Specifically, we leverage SigLip (Zhai et al., 2023) to encode the part names from the dataset into semantic vectors. These semantic vectors are then used as targets for aligning point features through InfoNCE Loss (van den Oord et al., 2019), which encourages the model to map point-level features to their corresponding semantic representations.

Let $m$ represent the number of distinct part names in the current object category, denoted as $\{s_1, s_2, \cdots, s_m\}$. These part names are encoded using the SigLIP text encoder to obtain semantic representations $\boldsymbol{x}_k = SigLip(s_k), k = 1...m$. Given a set of $N$ feature/label pairs $\{\boldsymbol{f}_k, a_k\}_{k=1...N}$, where $a_k \in 1, \ldots, m$ is the index of the ground-truth part name for the $k$-th point feature, just like above. The semantic Loss is defined as:

$$\mathcal{L}_{Sem} = \sum_{i=1}^{N} -\log \frac{\exp\left(\boldsymbol{f}_i \cdot \boldsymbol{x}_{a_i} / \tau\right)}{\sum_{k=1}^{m} \exp\left(\boldsymbol{f}_i \cdot \boldsymbol{x}_k / \tau\right)} \tag{2}$$

where $\tau \in \mathbb{R}^+$ denotes the balancing coefficient in InfoNCE.

The total loss used for training combines both the geometric loss and the semantic loss:

$$\mathcal{L}_{total} = \mathcal{L}_{Geo} + \mathcal{L}_{Sem} \tag{3}$$

This combined loss function ensures that the model learns features that are both geometrically consistent within parts and semantically aligned with their corresponding part names. To further refine the feature representations, we propose a lightweight feature refinement network, consisting of a shallow per-point MLP. This network processes the output from the Sonata model and uses the total loss function to guide the learning process, as shown in Fig 2.

## 3.2 PART-AWARE DIFFUSION POLICY

**Problem Formulation.** Part-Aware Diffusion Policy is a diffusion-based model for action generation (Chi et al., 2023; Tie et al., 2025) that takes 3D point cloud observations and robot proprioceptive state as input to predict future action sequences (action chunks). We formulate visuomotor control as modeling the conditional distribution $p(A_t \mid \mathbf{o}_t)$, where the observation at time $t$ is $\mathbf{o}_t = [\mathbf{P}_t^1, \ldots, \mathbf{P}_t^n, \mathbf{q}_t]$, with $\mathbf{P}_t^i$ representing the point cloud from the $i^{\text{th}}$ camera view and $\mathbf{q}_t$ the proprioceptive state. The predicted action chunk is $A_t = [a_t, a_{t+1}, \ldots, a_{t+H-1}]$. We train a Denoising Diffusion Probabilistic Model (DDPM) (Ho et al., 2020) and use Denoising Diffusion Implicit Model (DDIM) (Song et al., 2022) for accelerated inference sampling. The denoising process is defined as:

$$\mathbf{a}_t^{k-1} = \frac{\sqrt{\bar{\beta}^{k-1}} \gamma^k}{1 - \bar{\beta}^k} \mathbf{a}_t^0 + \frac{\sqrt{\bar{\beta}^k}(1 - \bar{\beta}^{k-1})}{1 - \bar{\beta}^k} \mathbf{a}_t^k + \tau^k \mathbf{v}, \tag{4}$$

where $\{\beta^k\}_{k=1}^K$ and $\{\tau^k\}_{k=1}^K$ are scalar coefficients from a predefined noise schedule, $\gamma^k := 1 - \beta^k$, and $\bar{\beta}^{k-1} := \prod_{i=1}^{k-1} \beta^i$. The noise term is $\mathbf{v} \sim \mathcal{N}(\mathbf{0}, \mathbf{I})$ when $k > 1$; otherwise, $\bar{\beta}^{k-1} = 1$ and $\mathbf{v} = \mathbf{0}$. The model is trained by minimizing the mean squared error (MSE) between the ground-truth action $\mathbf{a}_t$ and the model's prediction:

$$\mathcal{L}(\phi) := \text{MSE}\left(\mathbf{a}_t, D_{\boldsymbol{\theta}}\left(\mathbf{o}_t, \tilde{\mathbf{a}}_t, k\right)\right), \tag{5}$$

where

$$\tilde{\mathbf{a}}_t := \sqrt{\bar{\beta}^k} \mathbf{a}_t + \sqrt{1 - \bar{\beta}^k} \epsilon, \quad \epsilon \sim \mathcal{N}(\mathbf{0}, \mathbf{I}), \quad k \sim \text{Uniform}(\{1, \ldots, K\}). \tag{6}$$

**Policy Design.** We employ our Part-Aware 3D Feature Field (PA3FF) as a frozen backbone to extract point cloud embeddings. These embeddings are then fed into a trainable Transformer encoder to aggregate per-point features into a global representation. Since the features provided by our backbone are semantically meaningful, we use the semantic embedding of the task-critical part name as the CLS token to guide this aggregation. Next, we concatenate the resulting global scene feature with the agent's pos and pass it through a two-layer MLP to reduce its length, producing the encoder's final output. Conditioned on the compact representation, a diffusion action head outputs the robot action.

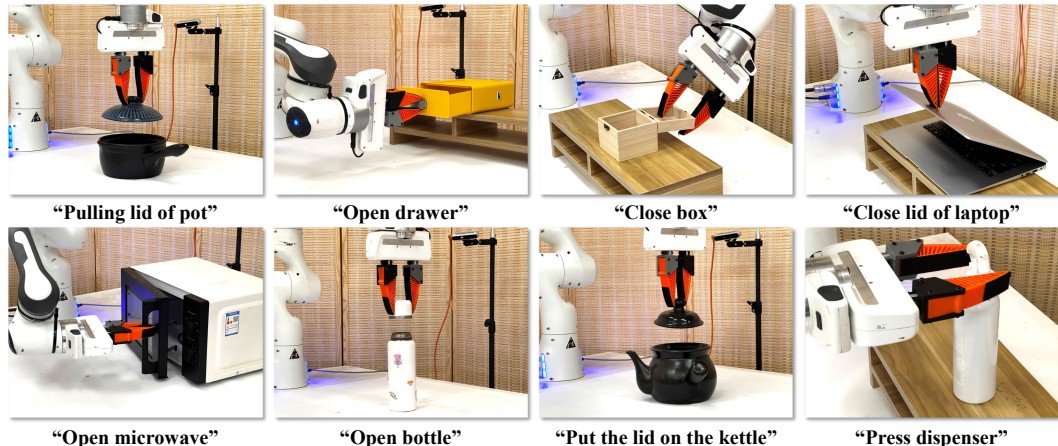

Figure 3: Task illustrations. We evaluate our model on eight downstream tasks.

# 4 EXPERIMENT

We systematically evaluate PA3FF and PADP through both simulation and real-world experiments, aiming to address the following questions: (1) How does the performance of our method compare to previous imitation approaches? (2) How well does our method generalize under object and environmental perturbations? (3) What factors contribute to the generalization of our method to novel instances? (4) Beyond imitation learning, what additional applications can PA3FF facilitate?

## 4.1 EXPERIMENTAL SETUP

We benchmark PADP in both simulated and real-world environments. The simulated environments serve as a controlled platform to ensure reproducible and fair comparisons. The real-world experiments demonstrate the method's applicability to real-world settings.

**Setup.** In simulation, we conduct multi-task training on the PartInstruct benchmark (Yin et al., 2025), which focuses on part-level fine-grained manipulation tasks. For real-world experiments, we use a Franka Emika Panda robotic arm equipped with UMI fingers (Chi et al., 2024) replacing the standard parallel gripper. Perception is handled by three Intel RealSense D415 depth cameras positioned around the workspace. Figure 9 illustrates our real-world setup and experimental objects.

**Tasks and Metrics.** We design 8 real-world tasks covering diverse manipulation scenarios (Figure 3). The Train split uses the same object instances/environment as demonstrations (with randomized initial poses); the Test split employs unseen objects/environments to evaluate Out-of-Distribution (OOD) generalization (Figure 5). Simulation adheres to PartInstruct's five-level generalization protocol (object states [OS], object instances [OI], task-type part combinations [TP], task categories [TC], object categories [OC]); see Appendix C.2 for details.

**Data Collections.** For the real-world experiments, we collect demonstrations by human teleoperation. The Franka arm and the gripper are teleoperated by the keyboard. Since our tasks contain more than one stage and include two robots and various objects, making the process of demonstration collection very time-consuming, we only provide 30 demonstrations for each task. For all six tasks, object poses are randomly initialized on the table. The action space contains the end-effector pose and gripper state, while observations include RGB images and corresponding depth images captured by three Intel RealSense D415 depth cameras. For simulation tasks, we leverage the demonstrations provided from Part-Instruct (Yin et al., 2025). More detailed settings can be found in Appendix E.

**Baselines.** In order to comprehensively evaluate PADP, we carefully select six baselines. These include image-to-action imitation learning baselines, *Diffusion Policy (DP)* (Chi et al., 2023). We also compare with models that have been specifically designed for 3D object manipulation including *Act3D* (Gervet et al., 2023), *RVT2* (Goyal et al., 2024), *3D Diffuser Actor (3D-DA)* (Ke et al., 2024b), *GenDP* (Wang et al., 2024c), *DP3* (Ze et al., 2024). These baselines show different 2D and 3D representations for policy. For DP, DP3, and GenDP, we add a language-conditioning module in the same manner as PADP to fuse language instructions. Details can be found in Appendix C.3

## 4.2 COMPARISON WITH BASELINES

Table 1: Simulated results across five test sets. The best-performing results are highlighted in bold

| Method | Test 1 (OS) | Test 2 (OI) | Test 3 (TP) | Test 4 (TC) | Test 5 (OC) | Average |
|---|---|---|---|---|---|---|
| Act 3D (Gervet et al., 2023) | $6.25_{\pm1.8}$ | $5.68_{\pm1.7}$ | $4.55_{\pm1.6}$ | $0.0$ | $2.08_{\pm2.1}$ | $3.88_{\pm1.8}$ |
| RVT2 (Goyal et al., 2024) | $4.55_{\pm2.0}$ | $4.55_{\pm2.0}$ | $6.36_{\pm2.3}$ | $0.91_{\pm0.9}$ | $3.33_{\pm3.3}$ | $4.04_{\pm2.1}$ |
| 3D-DA (Ke et al., 2024a) | $8.08_{\pm2.7}$ | $5.05_{\pm2.2}$ | $4.04_{\pm1.9}$ | $0.0$ | $3.70_{\pm3.6}$ | $4.26_{\pm1.0}$ |
| DP (Chi et al., 2023) | $7.27_{\pm1.8}$ | $8.64_{\pm1.9}$ | $8.18_{\pm1.8}$ | $3.75_{\pm2.1}$ | $6.67_{\pm3.2}$ | $5.96_{\pm2.2}$ |
| DP3 (Ze et al., 2024) | $23.18_{\pm2.8}$ | $23.18_{\pm2.8}$ | $18.18_{\pm2.6}$ | $7.73_{\pm1.8}$ | $6.67_{\pm3.2}$ | $15.40_{\pm2.6}$ |
| GenDP (Wang et al., 2024c) | $24.34_{\pm2.1}$ | $23.36_{\pm2.3}$ | $24.53_{\pm1.9}$ | $10.00_{\pm2.0}$ | $14.61_{\pm2.1}$ | $19.36_{\pm2.7}$ |
| **Ours** | $\mathbf{36.76_{\pm2.3}}$ | $\mathbf{34.33_{\pm3.6}}$ | $\mathbf{32.45_{\pm1.6}}$ | $\mathbf{13.75_{\pm2.0}}$ | $\mathbf{26.67_{\pm3.2}}$ | $\mathbf{28.79_{\pm2.5}}$ |

**Comparision PA3FF with other foundation features.** For an intuitive understanding of PA3FF, Figure 4 illustrates the feature fields of various objects. DINOv2 and SigLip, two 2D methods, use rendered images from 16 views of the mesh as input. After each image is encoded, the resulting features are remapped onto the mesh surface to generate a feature map. Sonata and our PA3FF, however, directly utilize point clouds sampled from the mesh as input to generate feature maps. The feature maps shown here for the 2D methods represent the result of mapping mesh features onto the point clouds input to Sonata and PA3FF. Compared to 2D foundation features like DINOv2 and SigLIP, our approach leverages the continuity of Sonata features, avoiding common **multi-view consistency** issues when aggregating 2D features from different views. As a result, the generated feature fields are smoother and less noisy (as shown in the faucet example). Our method also highlights key functional parts, such as microwave and refrigerator handles. Another limitation of 2D methods is their difficulty in **capturing small or thin parts**, which may occupy less than a patch or pixel in the 2D images and fail to be adequately represented (e.g., the refrigerator handle on the right). In contrast to Sonata, our approach explicitly promotes intra-part feature consistency and inter-part distinctiveness within specific object categories, leading to more semantically meaningful and discriminative part-level representations. More feature visualization can be found in Appendix A.3

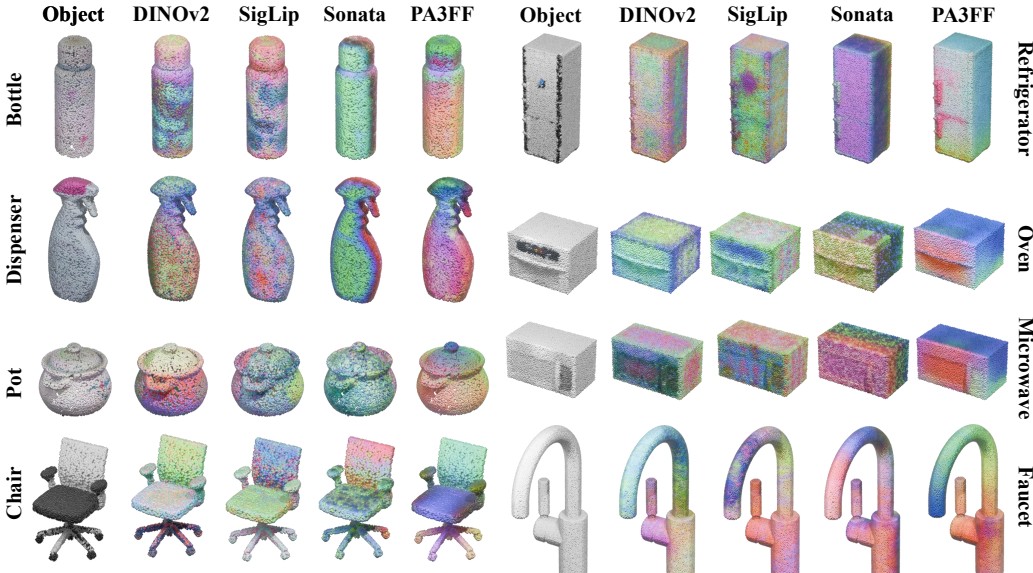

Figure 4: The feature field visualizations of PA3FF and other foundation features.

**Performance and Comparison.** Table 2 presents the main results on real-world tasks. PADP significantly outperforms several strong baselines across all tasks, by achieving a mean success rate of 58.75% under unseen objects, compared to the highest success rate of 35% achieved by baselines. Figure 9 presents snapshots of the real-world experiments. Table 1 shows the results on simulation tasks. Consistent with the real-world results, the simulation results also demonstrate that PADP enhances policy generalization.

Table 2: Real-world task success rates across different methods (train/test). Each task is evaluated with 10 trials under randomized initial conditions. The best-performing results are highlighted in bold

| Method/Task | Pulling lid of pot | | Open drawer | | Close box | | Close lid of laptop | |
|---|---|---|---|---|---|---|---|---|
| | *Train* | *Test* | *Train* | *Test* | *Train* | *Test* | *Train* | *Test* |
| DP (Chi et al., 2023) | 4/10 | 2/10 | 1/10 | 1/10 | 2/10 | 1/10 | 4/10 | 3/10 |
| DP3 (Ze et al., 2024) | 6/10 | 4/10 | 4/10 | 3/10 | 3/10 | 3/10 | 5/10 | 5/10 |
| GenDP (Wang et al., 2024c) | 7/10 | 6/10 | 5/10 | 5/10 | 3/10 | 3/10 | 6/10 | 4/10 |
| **PADP (Ours)** | **8/10** | **6/10** | **6/10** | **6/10** | **5/10** | **5/10** | **7/10** | **7/10** |
| Method/Task | Open microwave | | Open bottle | | Put lid on kettle | | Press dispenser | |
| | *Train* | *Test* | *Train* | *Test* | *Train* | *Test* | *Train* | *Test* |
| DP (Chi et al., 2023) | 1/10 | 0/10 | 5/10 | 2/10 | 1/10 | 0/10 | 0/10 | 0/10 |
| DP3 (Ze et al., 2024) | 3/10 | 1/10 | 4/10 | 3/10 | 3/10 | 1/10 | 1/10 | 1/10 |
| GenDP (Wang et al., 2024c) | 4/10 | 3/10 | 5/10 | 4/10 | 4/10 | 2/10 | 2/10 | 1/10 |
| **PADP (Ours)** | **6/10** | **5/10** | **8/10** | **6/10** | **6/10** | **5/10** | **4/10** | **3/10** |

Table 3: Generalization evaluation of the ***Open Bottle*** task (10 trials).

| Method | Completion Rate (%) | | | |
|---|---|---|---|---|
| | Original | Disturbance | | |
| | | Spatial | Object | Environmet |
| DP | 50 | 40 [↓10] | 10 [↓40] | 0 [↓50] |
| DP3 | 40 | 20 [↓20] | 20 [↓20] | 10 [↓30] |
| GenDP | 50 | 20 [↓30] | 30 [↓20] | 30 [↓20] |
| PADP (ours) | **80** | **70** [↓10] | **60** [↓20] | **60** [↓20] |

Following (Wang et al., 2025a), we evaluate generalization across three dimensions: *Spatial*, *Object* and *Environment*. **(1) Spatial Generalization:** When object poses change, baseline methods struggle to locate handles in the Open Microwave task. PADP consistently identifies correct grasping positions (Figure 15 in Appendix D) because its part-aware feature field effectively locates functional parts and understands object geometry. **(2) Object Generalization:** As demonstrated in Table 2, PADP maintains robust performance on unseen objects, while DP and DP3 fail on new instances. GenDP leverages semantic fields for improved generalization, but PADP surpasses all methods through PA3FF's precise feature representation. For example, in the Open Microwave task (see Figure 15 in Appendix D), Despite varying appearances across microwave models, PA3FF identifies shared functional structures (handles, bodies), enabling consistent manipulation regardless of shape or pose variations. **(3) Environment Generalization:** We evaluate robustness across four scenarios (Figure 5): original environment (*Situation 0*), added distractors (*Situation 1*), changed background (*Situation 2*), and combined changes (*Situation 3*). For the Open Bottle task, PADP maintains high performance across all conditions, while baselines degrade significantly in complex scenarios (Table 3). DP fails with background changes due to image dependency, DP3 resists color changes but struggles with distractors, and GenDP, despite outperforming other baselines, cannot handle the combined challenges of *Situation 3*.

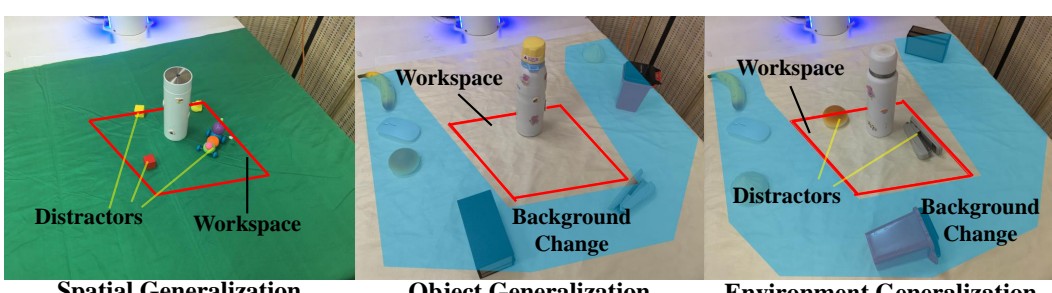

**Spatial Generalization**      **Object Generalization**      **Environment Generalization**

Figure 5: Generalization test set of the ***Open Bottle*** task.

### 4.3 Various Downstream Applications.

We evaluated the properties of the learned feature field in various applications, including part decomposition, 3D shape correspondences, and feature field consistency.

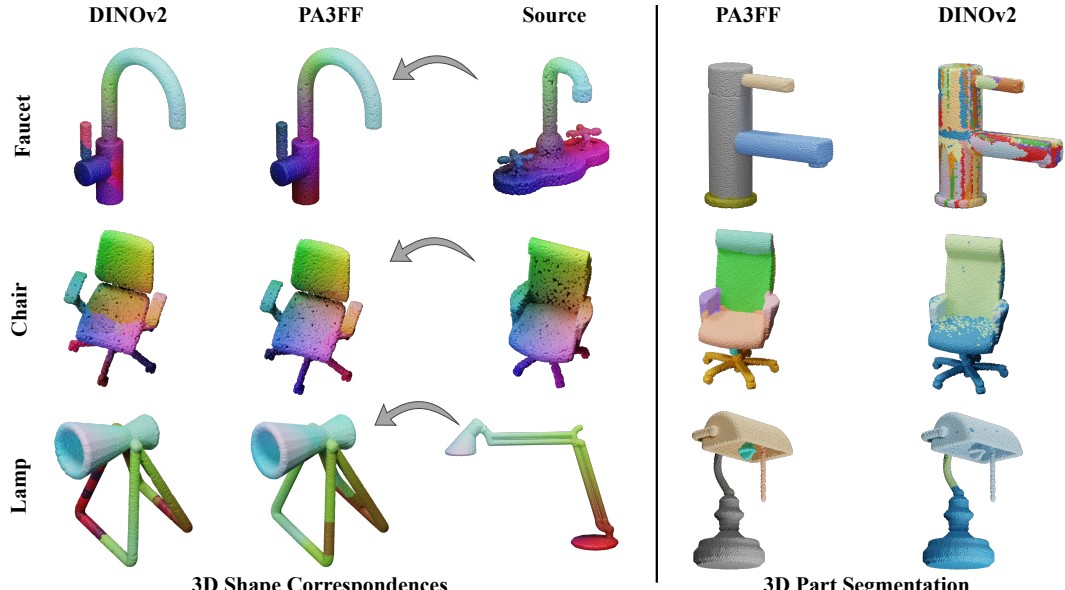

Figure 6: PA3FF exhibits consistency across shapes, enabling applications such as correspondence learning and part segmentation.

**3D Shape Correspondences.** The cross-shape consistency embedded in PA3FF provides a robust prior for fine-grained point-to-point correspondence learning. As a promising application, we use Functional Maps Ovsjanikov et al. (2012) to establish correspondences between a source and a target shape. To begin, we initialize the correspondences by finding nearest neighbors in the PA3FF feature space. These initial correspondences are then refined using Smooth Discrete Optimization Magnet et al. (2022), which iteratively solves functional maps in a coarse-to-fine fashion to recover a smooth and accurate point-to-point mapping. As shown in Figure 6, we compare the performance of this method against DINOv2 Oquab et al. (2023) features. In particular, PA3FF excels in providing precise correspondences, even in challenging cases where the shapes differ significantly in topology or pose. Beyond capturing shape similarity, PA3FF is capable of encoding functional semantics of parts—allowing it to match parts based on their function rather than just appearance—while still maintaining smoothness in the resulting correspondences.

**3D Part Segmentation.** PA3FF learns a part hierarchy implicitly through contrastive learning on diverse 3D data. This hierarchical structure can also be explicitly derived using agglomerative clustering. Figure 6 shows the results obtained from the clustering of the DINOv2 and PA3FF feature maps using identical parameters. Table 5 provide quantitative results of segmentation performance on the PartNet-Ensembled (PartNetE) (Liu et al., 2023) dataset. PA3FF successfully identifies significant relationships between parts, a feature that can be leveraged in a range of practical applications.

## 5 Conclusion

We propose Part-Aware 3D Feature Field (PA3FF), a novel 3D feature representation that enhances generalization for articulated object manipulation by focusing on functional parts. Combined with the Part-Aware Diffusion Policy (PADP), an imitation learning framework, PA3FF improves sample efficiency and generalization. Experimental results show that PADP outperforms existing 2D and 3D representations, including CLIP, DINOv2, and Grounded-SAM, in both simulated and real-world tasks. Beyond imitation learning, PA3FF enables a variety of downstream tasks, demonstrating its versatility and effectiveness in robotic manipulation.

ACKNOWLEDGMENT

We gratefully acknowledge the financial support provided by the National Natural Science Foundation of China [grant number 62376006]. This funding has been instrumental in enabling our research and the successful completion of this work.

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

## A    DETAILS OF PA3FF

Table 4: Comparisons with Other Foundation Features

| Method | Part-Aware | 3D Representation | Dense | Semantic |
|---|---|---|---|---|
| DINOv2 (Oquab et al., 2023) | ✗ | ✗ | ✗ | ✗ |
| SigLip (Zhai et al., 2023) | ✗ | ✗ | ✗ | ✓ |
| CLIP (Radford et al., 2021) | ✗ | ✗ | ✗ | ✓ |
| ULIP (Xue et al., 2023) | ✗ | ✓ | ✗ | ✓ |
| NDF (Simeonov et al., 2021) | ✗ | ✓ | ✓ | ✗ |
| D3Field (Wang et al., 2024d) | ✗ | ✓ | ✗ | ✓ |
| LERF (Kerr et al., 2023a) | ✗ | ✓ | ✗ | ✓ |
| Sonata (Wu et al., 2025b) | ✗ | ✓ | ✓ | ✗ |
| **PA3FF(Ours)** | ✓ | ✓ | ✓ | ✓ |

### A.1    COMPARISONS WITH OTHER FOUNDATION FEATURES

A key to generalization in articulated object manipulation lies in understanding *functional parts* (e.g., door handles, drawer knobs), which indicate *where* and *how* to manipulate across diverse object categories and shapes. As shown in Table 4, existing methods fall short in some critical dimensions:

**Semantic-only methods (CLIP, SigLip)** provide global semantic understanding but lack 3D geometric grounding and dense spatial resolution. They cannot answer "which specific point on this handle should the robot grasp?"

**3D-aware methods (ULIP, LERF, D3Field)** lift 2D features into 3D space but remain *coarse-grained* and *non-part-aware*. They struggle with within-part geometric coherence: a handle on a round door versus one on a rectangular cabinet should share feature similarity despite shape differences.

**Dense geometric methods (NDF, Sonata)** achieve dense 3D representations but lack *semantic separability*. They cannot distinguish a handle from a knob based on functional semantics—only on geometric similarity.

**Our Solution: Part-Aware 3D Feature Fields.** We propose **PA3FF**, the first feature representation that simultaneously satisfies all four critical properties (Table 4): part-aware, 3D-native, dense, and semantically grounded. This is *essential* for manipulation: a robot must recognize that a point is not merely "part of the object" but specifically "the handle," consistently across unseen, topologically distinct objects. PA3FF uniquely addresses this gap, enabling true part-level generalization for the manipulation of articulated objects.

### A.2    LIMITATIONS OF FEATURE LIFTING UP

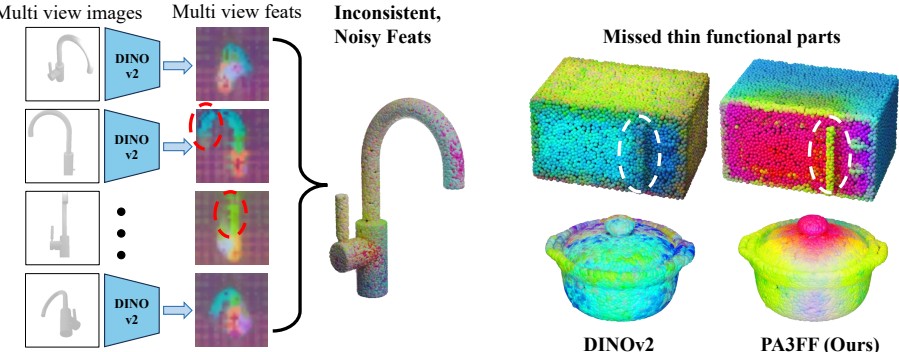

Figure 7: Flaws of lifting up method.

Although 3D priors are known to enhance generalization, lifting features from 2D to 3D introduces significant challenges. Models that naively average multi-view features from frozen 2D networks suffer from inconsistent visibility across views. Rendered 2D images can also miss thin or small parts like handles or buttons.

## A.3 MORE FEATURES' VISUALIZATIONS

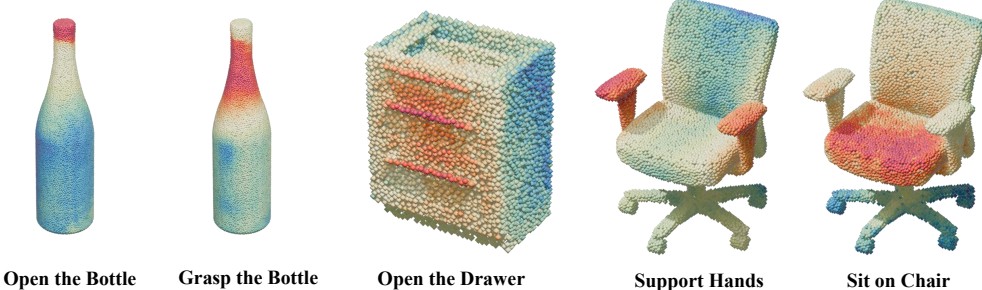

| Open the Bottle | Grasp the Bottle | Open the Drawer | Support Hands | Sit on Chair |

Figure 8: Heatmaps of point cloud features' similarity with several queries' text encoding.

Since the features generated by PA3FF contain semantic information, calculating feature similarity using different task statements for the same object allows us to focus on different parts of the object. The figure shows the heatmap visualization of cosine similarity between different encoded instructions and features.

## A.4 MORE QUANTITATIVE RESULTS OF PA3FF

Table 5: **Segmentation Results on the PartNetE Dataset.** Category mAP50s (%) are shown for different object categories. Higher values indicate better performance.

| Method | Bottle | Chair | Display | Lamp | Storage Furniture | Table | Average |
|---|---|---|---|---|---|---|---|
| PointGroup Jiang et al. (2020) | 8.0 | 77.2 | 16.7 | 9.8 | 0.0 | 0.0 | 18.6 |
| SoftGroup Vu et al. (2022) | 22.4 | 87.7 | 27.5 | 19.4 | 11.6 | 14.2 | 30.5 |
| PartSlip Liu et al. (2023) | 79.4 | 84.4 | 82.9 | 68.3 | 32.8 | 32.3 | 63.4 |
| PartSlip++ Zhou et al. (2023) | 78.5 | 86.0 | 74.1 | 66.9 | 36.7 | 33.5 | 62.6 |
| **Ours** | **94.6** | **90.0** | **86.5** | **69.5** | **49.6** | **33.4** | **70.6** |

# B    ABLATION STUDY

To validate the effectiveness of our proposed PADP framework, we conduct comprehensive ablation studies on the individual components of our method. Each ablation targets a specific design choice to understand its contribution to the overall performance.

## B.1    ABLATION COMPONENTS

We evaluate the following key components of our approach: **Stacking Additional Transformers:** We modify the PTv3 architecture by removing most downsampling layers and instead deepen the network through stacking additional transformer blocks. This architectural change enhances detail preservation while improving feature abstraction capabilities.

**Feature Refinement via Contrastive Learning:** We apply feature refinement through contrastive learning across different objects(Geometric & Semantic) to enhance part-level consistency and distinctiveness in the learned representations.

Table 6: Ablation study results showing the impact of different components on task performance across diverse tasks.

| Method | Put in Drawer (%) | Turn Tap (%) | Open Box (%) |
|---|---|---|---|
| PADP (Full Method) | **62** | **69** | **66** |
| w/o Stacking Additional Transformers | 58 | 63 | 59 |
| w/o Geometric Loss | 54 | 57 | 55 |
| w/o Semantic Loss | 46 | 53 | 52 |
| Sonata + DP3 | 39 | 50 | 44 |
| DP3 (Baseline) | 37 | 47 | 39 |

## B.2    ANALYSIS AND DISCUSSION

Table 6 presents the quantitative results of our ablation study on the "Put in Drawer" task, demonstrating the contribution of each component to the overall performance. Our ablation study reveals several critical insights into the effectiveness of PADP:

**Limited Gains from Direct Combination:** The combination of Sonata with DP3 achieves only 39% success rate, representing a modest improvement of 2% over the DP3 baseline (37%). This demonstrates that simply integrating existing methods without architectural modifications yields limited performance gains.

**Impact of Architectural Modifications:** Removing the feature refinement component while maintaining our transformer modifications results in 46% success rate. This 7% improvement over the baseline combination indicates that our architectural changes to the transformer stack provide meaningful benefits for manipulation tasks.

**Critical Role of Feature Refinement:** The most substantial performance degradation occurs when removing the feature refinement component, dropping from 62% to 46%. This 16% decrease highlights the importance of contrastive learning for achieving part-level consistency and distinctiveness in the learned representations.

**Synergistic Effect of Components:** The full PADP method achieves 62% success rate, demonstrating that the combination of architectural modifications and feature refinement creates a synergistic effect that significantly outperforms individual components.

These findings underscore that PADP's performance gains stem primarily from our novel part-aware feature field learning approach rather than merely utilizing Sonata representations. The substantial improvement from 46% to 62% when including feature refinement confirms that our algorithmic contributions are essential for addressing part-level manipulation challenges, and that Sonata alone is insufficient for solving these complex robotic tasks.

## C    Experiment Setup

In this section, we provide a detailed description of the experimental setup, including both real-world and simulation configurations, as well as a discussion of the baselines.

### C.1    Real-world Environment Setup

#### C.1.1    Hardware setup

For the real-world experiments, our experimental platform is built around a Franka Emika Panda robotic arm, with its parallel gripper's fingers replaced by UMI fingers (Chi et al., 2024). For perception, we employ three Intel RealSense D415 depth cameras. Figure 9 shows our real-world setup and object used.

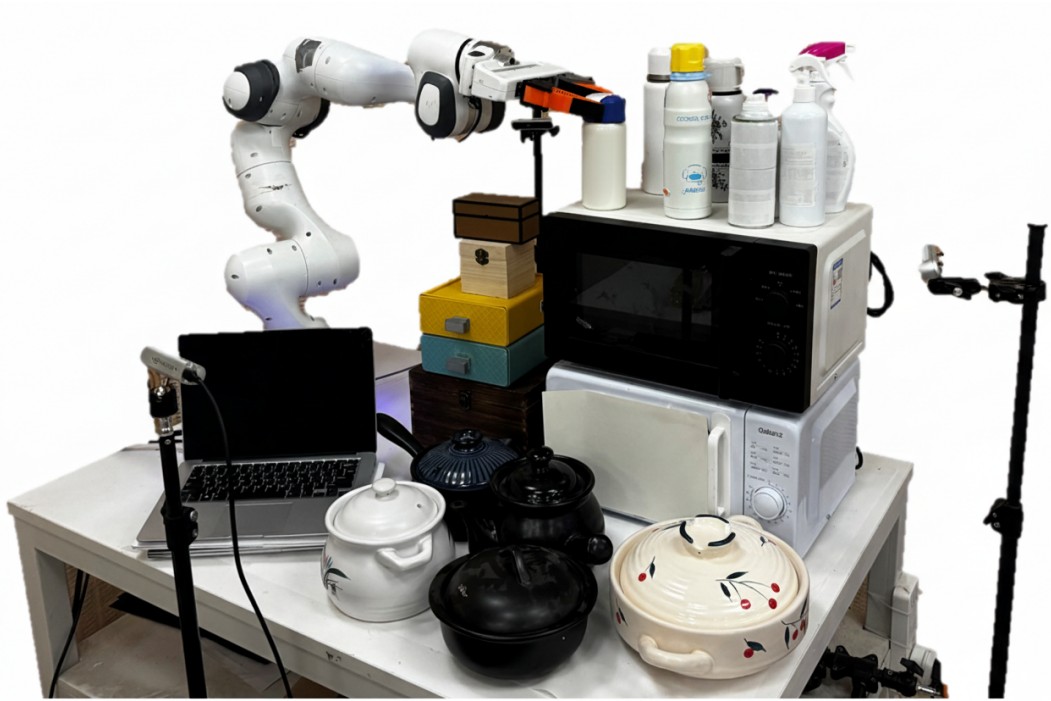

Figure 9: Real-world experiment environment and assets utilized.

#### C.1.2    Tasks Details.

For the real-world experiments, we selected 8 representative tasks, covering a variety of manipulation scenarios—including **pulling lid of pot**, **open drawer**, **close box**, **close lid of laptop**, **open microwave**, **open bottle**, **put the lid on the kettle**, and **press dispenser** to evaluate our system's performance across diverse challenges, as shown in Figure 3.

- **Pulling lid of pot** : A pot with a detachable lid is placed on the workspace. The robot must accurately grasp the lid's handle, lift it vertically, and separate it from the pot.

- **Open drawer**:A small drawer unit with a front handle is present. The robot must grasp the handle and smoothly pull the drawer forward until it is fully open.

- **Close box**: A small hinged box sits on the table. The robot must grasp the lid's edge and rotate it along its hinge axis until the box is fully closed.

- **Close lid of laptop**:An open laptop lies on the workspace. The robot must grasp the top edge of the screen and fold it down smoothly until the laptop is fully closed.

- **Open microwave**:A microwave oven with a front-mounted handle is present. The robot must grasp the handle and pull the door outward until it is fully open.
- **Open bottle**:A bottle with a screw-on cap stands on the table. The robot must grasp the cap, align its threads with the bottle neck, rotate it in the instructed direction by the required angle until the cap is fully unscrewed, and then lift the cap off vertically.
- **Put the lid on the kettle**:A kettle with a removable lid is placed on the workspace. The robot must grasp the lid, align it with the kettle opening, place it smoothly on top, and press down vertically until it is securely seated.
- **Press dispenser**:A dispenser bottle with a squeeze nozzle is placed on the workspace. The robot must grasp the nozzle section with its parallel gripper and apply a squeezing force to actuate the nozzle.

## C.2 SIMULATION ENVIRONMENT SETUP

Table 7: Summary of the five test sets and the type of generalization each addresses.

| Test Set | Type of Generalization |
|---|---|
| Test 1 (OS) | Novel object positions and rotations |
| Test 2 (OI) | Novel object instances within the same category |
| Test 3 (TP) | Novel part combinations within the same task categories |
| Test 4 (TC) | Novel part-level manipulation task categories |
| Test 5 (OC) | Novel object categories |

**Benchmarks.** PartInstruct contains 513 object instances across 14 categories (each annotated with part-level information) and 1302 fine-grained manipulation tasks grouped into 16 task classes. These 16 task classes include 10 seen categories for training and 6 unseen categories for testing, with each category defined by tasks that require the robot to perform a specific combination or sequence of part-level interactions.

**Evaluation.** To systematically evaluate the performance of the learned policy, PartInstruct designed a five-level evaluation protocol (see Table 7). Each test set evaluates a policy in one type of generalization condition. Specifically, they focus on generalizability over initial **object states(OS)** , novel **object instances (OI)**, novel part combinations in the same **task type** (TP), novel **task categories** (TC), and novel **object categories** (OC). Detailed visualization can be viewed in Figure 10, 11, 12, 13, 14.

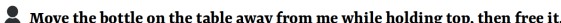

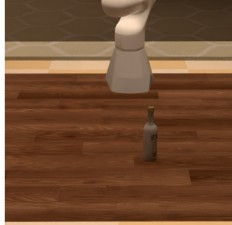 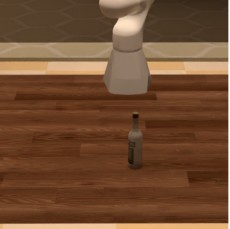

Figure 10: Left: Training set. Right: Test 1(OS). Novel object positions and rotations

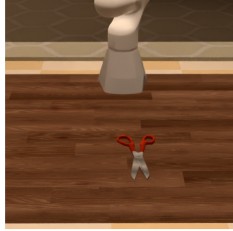 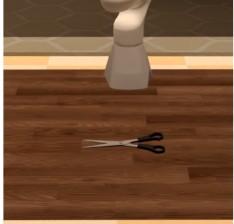

Figure 11: Left: Training set. Right: Test 2(OI). Novel object instances within the same category

👤 **Grab the top of the mug and move it forwards.**

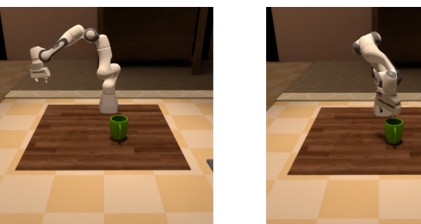
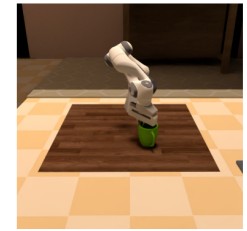

👤 **Grab the handle of the mug and move it backwards.**

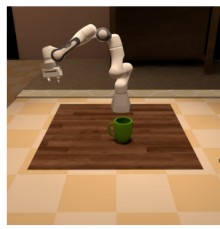
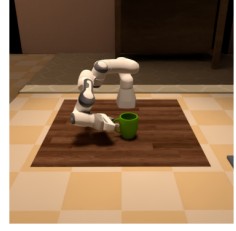
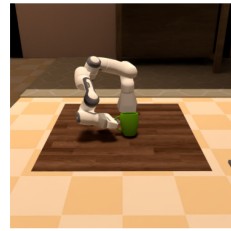

Figure 12: Above: Training set. Below: Test 3(TP). Novel part combinations within the same task categories

👤 **Lift the bucket by its left, then rotate the left part to face front, then move it to the left.**

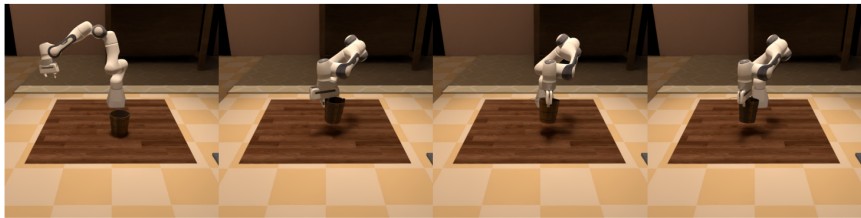

👤 **Move the bucket towards to the left in the air, then rotate handle to point towards back.**

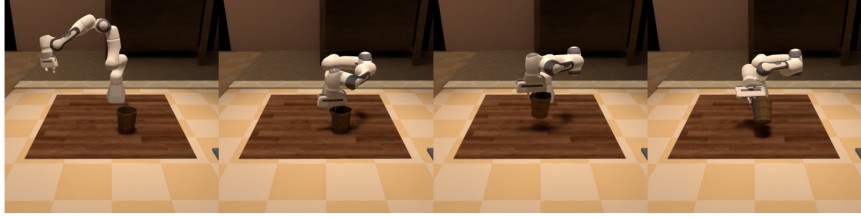

Figure 13: Above: Training set. Below: Test 4(TC). Novel part-level manipulation task categories

👤 **Lift the bottle by its top.**  👤 **Lift the knife by its base body.**

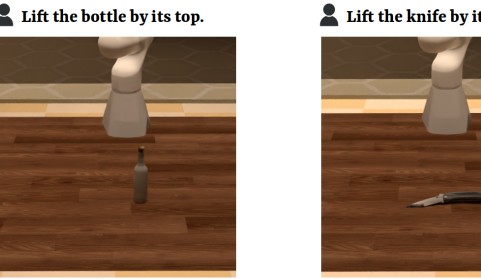

Figure 14: Left: Training set. Right: Test 5(OC). Novel object categories

### C.3 DETAILS OF BASELINES

**Diffusion Policy (DP)** We train a CNN-based DP from scratch; the action prediction horizon is set to 16 steps, with an observation horizon of 2 steps and action steps of 8. The input RGB images are cropped to a size of $76 \times 76$. For language instructions, we use a pre-trained T5-small language encoder to obtain a language embedding of 512 dimensions. This language embedding is then concatenated with other features to form the final feature representation.

**GenDP** Following the raw work (Wang et al., 2024c), we use DINOv2 and Grounding-DINO to extract information from multi-view 2D RGB images, and then project arbitrary 3D coordinates back to each camera, interpolate to compute representations from each view, and fuse these data to derive the descriptors associated with these 3D positions.

**3D Diffusion Policy (DP3)** The DP3 model is trained under a similar setup as DP, with an action prediction horizon of 16 steps, an observation horizon of 2 steps, and action steps of 8. For the point cloud observations, we use an input size of 1024 points, which are downsampled from the original point cloud using the Iterative Farthest Point Sampling algorithm (Qi et al., 2017). The language instructions are processed in DP3 following the same approach as in DP.

**Act3D** Act3D takes an image input size of $256 \times 256$. The action prediction horizon is set to 6 steps, and the observation horizon is 1 step. Following the raw work Gervet et al. (2023), we use ResNet50He et al. (2016) as the vision encoder, and use CLIP Radford et al. (2021) embeddings for vision-language alignment. For 3D action map generation, the number of "ghost" points is set to be 10,000, with a number of sampling level of 3.

**3D Diffuser Actor (3D-DA)** For 3D-DA, we use the front-view RGB and scene point cloud as vision inputs. The RGB image has a resolution of $256 \times 256$. Following Ke et al. (2024a), we extract visual features with a pre-trained CLIP ResNet-50 encoder and use CLIP Radford et al. (2021) embeddings for vision–language alignment. We use an interpolation length of 5 steps and an observation history of 3 steps.

**RVT2** We first convert the depth map from the static camera view into a point cloud in the camera coordinates, then apply camera extrinsic to transfer the point cloud into the world coordinates, where the action heat maps will be generated, and apply supervision. The action prediction horizon is chosen to be 6 steps, and the observation horizon is set to be 1 step.

## D MORE REAL-WORLD EXPERIMENT RESULTS

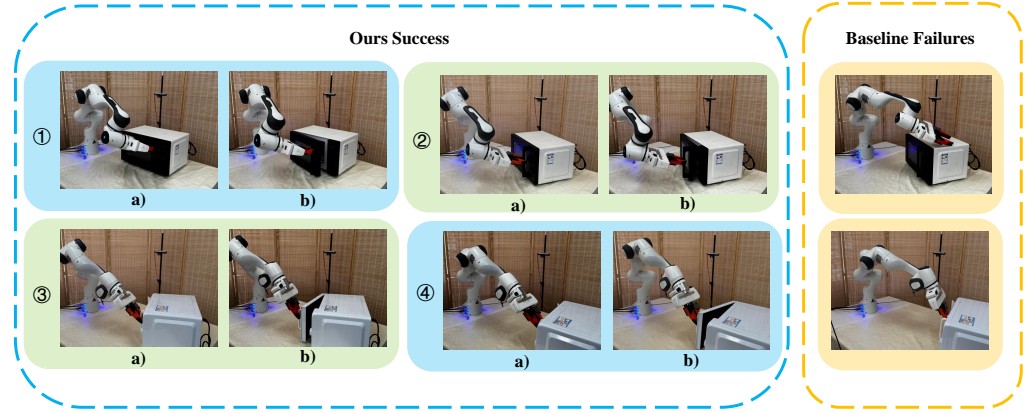

Figure 15: Our policy reliably detects variations in object parts across different positions and successfully executes the corresponding tasks, whereas the baseline method exhibits weaker spatial generalization and struggles to accurately perceive the object parts after positional changes.

1. **Spatial Generalization of PADP**: In the "Open microwave" task, our policy accurately perceives and localizes the door handle even when the microwave is spatially displaced,

allowing it to reliably grasp the handle and pull the door open (shown in Figure 15 left). In contrast, the baseline method exhibits weaker spatial generalization: it fails to correctly identify the microwave and its handle after translation or rotation, often grasping incorrect regions and failing to open the door (shown in Figure 15 right).

2. **Environment generalization of PADP**: Other than the "Open microwave" task, we also conducted environment generalization experiments on the "Open bottle" task. As shown in Table 3 and Figure 5, our policy maintains robust performance across varied spatial configurations, whereas baseline methods suffer a significant drop in success rate.

# E   MORE SIMULATION RESULT

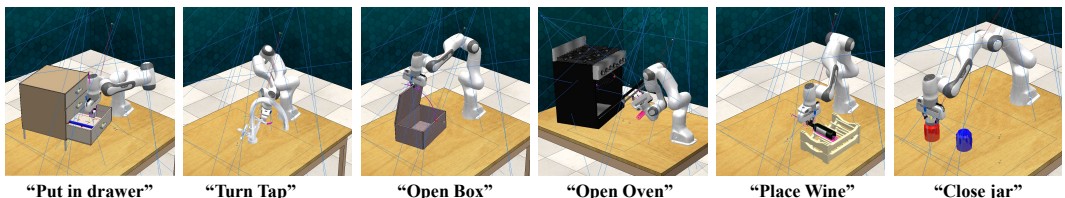

| "Put in drawer" | "Turn Tap" | "Open Box" | "Open Oven" | "Place Wine" | "Close jar" |

Figure 16: Visualizations of simulation environments.

**Benchmarks.** RLBench comprises 100 completely unique, hand-designed tasks performed using a 7-DoF Franka Emika Panda arm with a parallel gripper. Observations include rich proprioceptive data (joint angles, velocities, forces, gripper state) and visual data (RGB, depth, segmentation masks) from an over-the-shoulder stereo camera and an eye-in-hand monocular camera. Each task provides an infinite supply of motion-planned demonstrations, supporting research in imitation and few-shot learning.

**Tasks Details.** For our simulation experiments, we select 6 tasks from the RLBench, as shown in Figure 16.

- **Put item in drawer** : In this scenario, a cabinet features three drawers—upper, middle, and lower—with a small cube resting on top. The objective is for the robot to grasp the bottom drawer's handle, pull it open to an accessible position, and then precisely insert the cube into the opened drawer.

- **Turn tap**:In this scenario, two taps—one on the left and one on the right—each equipped with a rotary handle, are present. The objective is for the robot to interpret grasp that left tap's handle, and rotate it in the instructed direction until the tap is fully turned on.

- **Open box**: In this scenario, a box with a hinged lid is placed before the robot. The objective is to accurately grasp the edge of the lid and lift it along its hinge axis until the box is fully open.

- **Open oven**:In this scenario, an oven with a front-facing handle is present. The objective is for the robot to firmly grasp the oven door handle and pull it outward along its hinge until the oven is fully open.

- **Stack wine**:In this scenario, a standard wine bottle and a horizontal-slot wine rack—designed to cradle bottles on their sides—are placed on the work surface. The objective is for the robot to securely grasp the bottle, slide it into the designated rack slot on its side, and ensure it is evenly supported and perfectly centered without tilting.

- **Close jar**:In this scenario, two jars and a single lid are presented. The robot must grasp the lid, align its threads with the jar on the left, and rotate it in the instructed direction until the lid is fully tightened and sealed.

**Training Details.** We collected expert demonstrations using the built-in scripted policies in RLBench, gathering 25 and 50 demonstrations per task for training. For DP and DP3, after training for at least **2000** epochs, we select the single best-performing epoch for evaluation: we run each policy over 10 episodes, choose the top three success rates, and report their average as the final performance metric.

For our policy, we instead select the best-performing epoch within the range of epochs **300** to **400** and apply the same evaluation protocol.

**Results.** We evaluate our method on 6 RLBench tasks, as shown in Figure 16 and Table 8. Consistent with the real-world results, the simulation results also demonstrate that PADP enhances learning efficiency and generalization.

Table 8: Simulated results with different numbers of demonstrations. Here we use the two seen objects in the training phase to test the success rate, and conduct five trials for each object with **random initialization** in each task.

| Method/Task | *Put in Drawer* | | *Turn Tap* | | *Open Box* | | *Open Oven* | | *Place Wine* | | *Close Jar* | |
| Number of Demos | 25 | 50 | 25 | 50 | 25 | 50 | 25 | 50 | 25 | 50 | 25 | 50 |
|---|---|---|---|---|---|---|---|---|---|---|---|---|
| DP | 16% | 22% | 19% | 26% | 13% | 18% | 23% | 31% | 11% | 16% | 4% | 7% |
| DP3 | 31% | 37% | 41% | 47% | 34% | 39% | 51% | 59% | 41% | 49% | 9% | 13% |
| GenDP | 38% | 44% | 48% | 55% | 41% | 48% | 63% | 68% | 48% | 56% | 14% | 19% |
| PADP (Ours) | **52%** | **62%** | **61%** | **69%** | **56%** | **66%** | **77%** | **86%** | **63%** | **74%** | **22%** | **30%** |

## F  LIMITATION

Our work focuses primarily on rigid and articulated objects. While it can be extended to handle flexible objects with relatively simple deformations—such as cables or ropes—it remains limited when applied to deformable objects exhibiting complex structural changes after deformation. The foundation model we rely on, Sonata, struggles to cope with such challenging cases. Addressing objects with complex deformations remains an open problem for future research. We hope our work can inspire further exploration in this direction.

## G  THE USE OF LARGE LANGUAGE MODELS

We used a Large Language Model (LLM) only as a writing assistant to polish the language of the manuscript (*e.g.*, grammar refinement, style adjustment, and clarity improvement). The research ideas, methodology design, experiments, and analysis were entirely conceived, implemented, and validated by the authors without reliance on the LLM. The LLM did not contribute to research ideation, experimental design, or result interpretation.

