# OpenReview forum: "PA3FF:Learning Part-Aware Dense 3D Feature Field For Generalizable Articulated Object Manipulation"
_ICLR.cc/2026/Conference — ICLR 2026 Poster_

### Official Review · Reviewer_SDVB · 2025-10-30

**Soundness:** 2
**Presentation:** 3
**Contribution:** 3
**Rating:** 6
**Confidence:** 4

**Summary:**

This paper argues that robots handle new objects better when they reason about the parts that matter for action—like handles, buttons, and lids—rather than whole objects. The authors introduce a 3D “part-aware” feature field that turns a point cloud into dense features where points on the same functional part look alike, and those features are tied to plain-language part names. They then use this representation in a diffusion policy that conditions control on the named part, letting the robot plan motions directly from the 3D scene. Because the features are native to 3D, they’re more consistent across viewpoints and make part boundaries clearer. In experiments spanning simulation and eight real-world tasks, the method outperforms strong 2D-feature and 3D-policy baselines, particularly when generalizing to unseen objects and states. The same features also enable point-to-point correspondence and unsupervised part segmentation, making the approach a broadly useful backbone for part-centric perception and manipulation.

**Strengths:**

The paper introduces a 3D-native, part-aware representation that’s aligned with language and plugs cleanly into a diffusion control policy, leading to strong generalization across unseen objects, states, and tasks. The evaluation is thorough—covering simulation and eight real-world tasks with clear five-way generalization splits—and shows sizable gains over both 2D-lifted and 3D baselines. Beyond control, the same features enable point correspondence and unsupervised part segmentation, and ablations clarify why the 3D-native design outperforms view-lifted alternatives.

**Weaknesses:**

1. The work has extensive evaluation on robot experiment, but lack of quantitative evidence on the feature field quality.
2. Runtime/latency: PADP runs at ~4.23 FPS vs. DP/DP3 at ~12 FPS, limiting high-frequency control.
3. Dependence on part supervision & external text embeddings: training leans on labeled parts and SigLIP part-name embeddings; baselines may not use comparable supervision.

**Questions:**

Here are more feature splatting paper to cite:
LERF: Language Embedded Radiance Fields (ICCV 2023)
Feature Splatting: Language-Driven Physics-Based Scene Synthesis and Editing (ECCV 2024)
M3: 3D-Spatial Multimodal Memory (ICLR 2025)

---

> ### Author Response · Authors · 2025-11-26
>
> # Dear Reviewer SDVB
>
> We sincerely thank you for your thoughtful and constructive feedback. Your comments helped us better clarify the and limitations of our work, while providing more valuable perspectives on improving our work. We address each point below.
>
> > ## W1: Lack of quantitative evidence on the feature field quality
>
> As shown in Appendix A.4, We provide more quantitative results of PA3FF, these results demonstrate that the feature field learned by our method is expressive enough to support downstream tasks.
>
> ### Table 5: Segmentation results on the PartNetE dataset. Category mAP50s (%) are shown.
>
> | Method       | bottle | chair | display | lamp  | storage furniture | table |
> |--------------|--------|-------|---------|-------|-------------------|-------|
> | PointGroup[1]   | 8.0    | 77.2  | 16.7    | 9.8   | 0.0               | 0.0   |
> | SoftGroup[2]    | 22.4   | 87.7  | 27.5    | 19.4  | 11.6              | 14.2  |
> | PartSlip [3]  | 79.4   | 84.4  | 82.9    | 68.3  | 32.8              | 32.3  |
> | PartSlip ++ [4]|78.5|86.0|74.1|66.9| 36.7| 33.5|
> | Ours         | 94.6   | 65.1  | 86.5    | 69.5  | 49.6              | 33.4  |
>
>
> > ## W2: Inference speed trade-off
>
> The $4.23$ FPS (~$236$ ms latency) is an acceptable and necessary trade-off for high-fidelity generalization in this domain. In the context of articulated object manipulation (e.g., opening a microwave), the robot moves relatively slowly to interact with physical constraints. This latency is comparable to human reaction time and is sufficient for closed-loop control in these scenarios. The trade-off is worthwhile because standard methods (like DP3) often fail completely on unseen object geometries (success rate < 20% on hard tasks). A 12.7 FPS policy that fails to grasp a novel handle is less useful than a 4.23 FPS policy that succeeds. We believe accurate perception is the bottleneck for generalization in this domain, making the computational cost acceptable. Notably, while our method is slower than DP3 approaches due to 3D processing overhead, it significantly outperforms GenDP and maintains practical real-time performance for robotic manipulation.
>
> > ## W3: Unfair comparsion with baselines
>
> Thank you for raising this important concern about fairness in our experimental comparison. We would like to clarify several key points regarding the baseline methods and our approach.
>
> **Pretrained Model Usage in Baselines**
>
> GenDP does utilize pretrained foundation models, specifically leveraging Grounded-SAM and DINOv2 to generate 3D descriptor fields from multi-view 2D features following D³Fields [1,2]. Therefore, our comparison is fair regarding pretrained visual representations.
>
> **Part Information as Core Contribution**
>
> The utilization of part-level information represents our core methodological contribution rather than an unfair advantage. When comparing with existing part-aware approaches [3,4,5], most involve per-shape optimization requiring multi-step inference pipelines (rendering, segmenting multiple views, then fusing into 3D), leading to lengthy runtimes and multi-view inconsistencies. In contrast, PA3FF is a feedforward model that achieving significantly faster inference compared to optimization-based methods.
>
> > ## Q1: More related works on feature splatting
>
> Thank you for suggesting these additional works. We have added these related work in the revised manuscript. While these approaches construct 3D feature fields by lifting 2D image features into 3D, our method operates directly on native 3D point clouds, producing a spatially smooth and geometry-consistent feature field. We appreciate your helpful pointers, which improved the completeness of our related work discussion.
>
> ### **Reference**
>
> [1] Yixuan Wang, Mingtong Zhang, et al. D³Fields: Dynamic 3D Descriptor Fields for Zero-Shot Generalizable Rearrangement. In Conference on Robot Learning (CoRL), 2024.
>
> [2] Yixuan Wang, Guang Yin, et al. GenDP: 3D Semantic Fields for Category-Level Generalizable Diffusion Policy. In Conference on Robot Learning (CoRL), 2024.
>
> [3] Haodi He, Colton Stearns, Adam W. Harley, and Leonidas J. Guibas. View-consistent hierarchical 3d segmentation using ultrametric feature fields. In Conference on Computer Vision and Pattern Recognition (CVPR), 2024.
>
> [4] Chung Min Kim, Mingxuan Wu, et al. Garfield: Group anything with radiance fields. In Conference on Computer Vision and Pattern Recognition (CVPR), 2024.
>
> [5] Yunhan Yang, Yukun Huang, et al. Sampart3d: Segment any part in 3d objects. arXiv preprint arXiv:2411.07184, 2024.
>
> ```
> Thank you once again for your valuable feedback. We hope our clarification has addressed your concerns. Please contact us if you have any further questions or concerns.
> ```

---

### Official Review · Reviewer_Kvwg · 2025-10-31

**Soundness:** 3
**Presentation:** 3
**Contribution:** 2
**Rating:** 6
**Confidence:** 4

**Summary:**

Summary:
The paper proposes PA3FF, a 3D-native, part-aware dense feature field learned directly from point clouds, and PADP, a diffusion policy that uses PA3FF as a frozen perception backbone with language and robot-state conditioning. Together they target articulated-object manipulation with better sample efficiency and generalization to unseen objects, showing superior results to prior 2D/3D features plus diffusion-policy baselines in both simulation and real-world tasks, and also enabling downstream uses like correspondence and part segmentation.

Main contributions:
1.PA3FF: a part-aware 3D feature field that enforces within-part feature coherence and between-part separability, trained with contrastive objectives and text alignment to functional part names.
2.PADP: a diffusion policy built on the frozen PA3FF backbone, conditioned on language cues and robot state, improving sample efficiency and cross-instance and cross-category generalization.
3.Strong empirical gains over representative baselines (e.g., CLIP/DINOv2 features and DP/GenDP families) across multiple generalization splits in simulation and eight real-world articulated-object tasks.
4.Versatility of the learned representation, supporting additional perception tasks such as shape correspondence and part segmentation.

**Strengths:**

### Quality

* Pipeline is reasonably complete: pretrained 3D backbone, contrastive representation learning, language conditioning, and diffusion policy, with corresponding ablations.
* Experiments cover simulation and a modest set of real tasks, include cross-instance/category splits, and compare against common 2D/3D feature and diffusion baselines with generally consistent gains.

### Clarity

* Problem framing is clear: focus on functional-part consistency to address articulated-object manipulation bottlenecks.
* Exposition is structured (representation → policy), with losses and inputs described in layers; implementation details are sufficient for high-level replication.

### Significance

* Potential to reduce instance-specific engineering and data needs, especially under shifts from unseen objects or deformations.

**Weaknesses:**

1. Insufficient novelty (core issue)
   The paradigm—“part-aware dense 3D representation + language prompts + diffusion policy”—reads as a combination/tuning of existing components (NDF-style dense correspondence, DP3/GenDP-style 3D-aware diffusion, ULIP-style language–3D alignment). The manuscript does not present an indispensable conceptual increment (new inductive bias/new representational property/new problem formulation); current differences are mainly in implementation and loss engineering.
   — Suggestion: Use a “conceptual comparison + ablation proof” to pinpoint your **single unique idea**: show that removing that idea (e.g., the part-consistency term or specific field structure) causes a **significant** drop, and provide head-to-head results against the strongest nearby baselines (DP3/GenDP/NDF variants).

2. Lacking verifiable “necessity evidence” for the representation claim
   You claim “part-aware” beats a “generic 3D semantic field,” but there is no counterfactual under matched supervision and budget to show the advantage comes from the representation itself rather than backbone scale or the pretraining data distribution.
   — Suggestion: Under the **same backbone and training budget**, swap only “part-aware field ↔ generic semantic field,” and report cross-task/cross-object gains with significance tests.

3. Unclear scope of the language module’s contribution
   Conditioning/alignment on part names is not novel, and there is no degradation/robustness quantification (synonyms, hierarchical terms, noisy/wrong labels, no-language variant) to show language is a **key** driver rather than a cosmetic add-on.
   — Suggestion: Provide curves of **language-noise strength → performance**, and report **per-task/per-part** marginal contributions.

4. Heavy reliance on strong pretrained backbones; insufficient factorized ablations
   A large portion of the gains may come from PTv3/Sonata-style pretraining; current ablations do not sufficiently disentangle backbone capacity from your objectives/structure.
   — Suggestion: Run **full-factorial ablations** (backbone type × with/without large-scale pretraining × with/without language alignment, contrastive losses, part supervision, structural tweaks).

**Questions:**

1. Please state the paper’s **single indispensable conceptual increment** (not implementation or loss details) and explain what **new inductive bias/representational property** it introduces.
2. Please provide a **conceptual comparison table** contrasting NDF / DP3 / GenDP / ULIP / *this work* , and mark which elements are **first introduced** by this paper.
3. Please report ablations that **remove the key new component(s)** (e.g., part-consistency loss, specific field structure, alignment mechanism): do all primary metrics **drop significantly**? Include statistical significance.
4. Under **identical sensing inputs, action space, number of demonstrations, and training budget**, run **head-to-head** comparisons against the strongest nearby baselines (DP3/GenDP/NDF variants). If the method still does not win, explain how the claimed “novelty” stands.
5. Please provide a **correlation analysis from field quality to performance**: e.g., within-part coherence, cross-view consistency versus success rate, with correlation coefficients and visualizations.
6. Please add **degradation and robustness** studies for the language component:
   * Synonym substitution (handle/knob/grip), hierarchical terms (door handle vs. handle);
7. How is the vocabulary constructed and disambiguated? How do you handle **same-name different parts** or **cross-category semantic drift**? Please provide **error cases and proportions**.
8. Please provide a **full-factorial ablation**:
   {Backbone: PTv3 / Point Transformer / others} ×
   {With/without Sonata or equivalent large-scale pretraining} ×
   Report results in **both simulation and real** settings.

**If the authors can satisfactorily address these questions, I would raise my score.**

---

> ### Comment · Reviewer_Kvwg · 2025-11-27
> **Official Comment by Reviewer Kvwg**
>
> I notice that the authors have responded to other reviewers but have not yet addressed any of my questions. I find it somewhat unprofessional and discourteous to respond to reviewers selectively rather than simultaneously. I am still waiting for clarification on several key points I raised in my review, particularly regarding the novelty of the approach and certain technical details that remain unclear to me. This is the only paper in my review queue where my concerns have not been addressed. I would appreciate a response before finalizing my evaluation, as these unresolved questions are critical for my assessment.

---

> > ### Author Response · Authors · 2025-11-27
> >
> > ## Dear Reviewer Kvwg,
> >
> > I sincerely apologize for the delay in responding to your questions, this was entirely unintentional. We have been fully focused on revising the manuscript in strict accordance with your requirements, which led to the temporary oversight in addressing your specific concerns promptly.
> >
> > We greatly value your feedback, and we have now thoroughly addressed all the key points you raised. Our responses are included in the revised manuscript for your reference. We hope our clarifications have effectively resolved your concerns. Please do not hesitate to contact us if you have any additional questions or need further elaboration on any aspect.
> >
> > Thank you again for your valuable input, which has helped improve our work significantly.
> >
> > Best regards,

---

> > > ### Comment · Reviewer_Kvwg · 2025-11-27
> > > **Official comment**
> > >
> > > Thank you for your comprehensive response. Your additional experiments and clarifications have addressed most of my concerns effectively. Based on these improvements, I am willing to raise my score.

---

> ### Author Response · Authors · 2025-11-27
>
> # Dear Reviewer Kvwg
>
> We sincerely thank the reviewer for the thorough and intellectually rigorous assessment and for acknowledging the quality of our pipeline, clarity, and strong empirical gains. We particularly appreciate the push for conceptual clarity and verifiable necessity evidence, which helps us position our work more precisely against strong related baselines. Below, we address the concerns and questions point-by-point.
>
> > ## W1 & Q1 & Q2 Comparison with other foundation feature
>
> **We have add a new seciton Appendix A.1 in the revised version.**
>
> A key to generalization lies in understanding functional parts (e.g., door handles and knobs), which indicate where and how to manipulate across diverse object categories and shapes.
>
> **Semantic-only methods (CLIP, SigLip)** provide global semantic understanding but lack 3D geometric grounding and dense spatial resolution. They cannot answer "which specific point on this handle should the robot grasp?"}
>
> **3D-aware methods (ULIP, LERF, D3Field)** lift 2D features into 3D space but remain **coarse-grained** and **non-part-aware**. They struggle with within-part geometric coherence: a handle on a round door versus one on a rectangular cabinet should share feature similarity despite shape differences.}
>
> **Dense geometric methods (NDF, Sonata)** achieve dense 3D representations but lack **semantic separability**. They cannot distinguish a handle from a knob based on functional semantics—only on geometric similarity.}
>
> **Our Solution: Part-Aware 3D Feature Fields.**
> We propose PA3FF, the first feature representation that simultaneously satisfies all four critical properties (Table 4): part-aware, 3D-native, dense, and semantically grounded. This is essential for manipulation: a robot must recognize that a point is not merely "part of the object" but specifically "the handle," consistently across unseen, topologically distinct objects. PA3FF uniquely addresses this gap, enabling true part-level generalization for the manipulation of articulated objects.}
>
> Table 4. Comparisons with Other Foundation Features
> | Method | Part-Aware | 3D Representation | Dense | Semantic |
> | :--- | :---: | :---: | :---: | :---: |
> | DINOv2 | **×** | **×** | **×** | **×** |
> | SigLip | **×** | **×** | **×** | **✓** |
> | CLIP | **×** | **×** | **×** | **✓** |
> | ULIP | **×** | **✓** | **×** | **✓** |
> | NDF | **×** | **✓** | **✓** | **×** |
> | D3Field | **×** | **✓** | **×** | **✓** |
> | LERF | **×** | **✓** | **×** | **✓** |
> | Sonata | **×** | **✓** | **✓** | **×** |
> | **PA3FF(Ours)** | **✓** | **✓** | **✓** | **✓** |
>
>
>
> The reviewer asks for a head-to-head comparison between a "part-aware field" and a "generic semantic field" under matched budget and backbone. This critical comparison is directly addressed by our ablation experiment(Table 6):
>
> **Table 7: Ablation study results showing the impact of different components on task performance across diverse tasks.**
>
> | Method | Put in Drawer (\%) | Turn Tap (\%) | Open Box (\%) |
> | :--- | :---: | :---: | :---: |
> | PADP (Full Method) | **62** | **69** | **66** |
> |   w/o Stacking Additional Transformers | 58 | 63 | 59 |
> |   w/o Geometric Loss| 54 | 57 | 55 |
> |   w/o Semantic Loss | 46 | 53 | 52 |
> | Sonata + DP3 | 39 | 50 | 44 |
> | DP3 (Baseline) | 37 | 47 | 39 |

---

> ### Author Response · Authors · 2025-11-27
>
> > ## W2, Q3, Q5, Q8 Ablation
>
> The reviewer asks if all primary metrics drop significantly when key new components are removed. This necessity is systematically proven in Table 6:
>
> **Modifications:** Standard PTv3’s aggressive downsampling was preventing dense feature extraction on small, critical parts (e.g., handles). We replaced the downsampling layers with Linear layers to preserve high spatial resolution. The subsequent stacking of Transformer blocks (encoder depths from $(2, 2, 2, 6, 2)$ to $(3, 3, 3, 6, 16)$) was a necessary engineering change to compensate for the increased receptive field requirement at the higher resolution.
>
> **Performance Analysis:**
> **Necessity of Geometric Loss:** Removing this loss leads to a performance drop from 58% to 54%, 63% to 57%, 59% to 55%. This validates the necessity of our Local Coherence Inductive Bias, which ensures geometrically consistent feature clustering for physical interaction.
>
> **Necessity of Semantic Loss:** We further remove the semantic loss resulting in a drop from 54% to 46%, 57% to 53%, 55% to 52%. This confirms the critical role of Language-Feature Alignment in enabling robust generalization across object categories.
>
> While the contrastive feature refinement is the primary driver (16% gain), the architectural change is the necessary foundation for high-resolution, dense feature fields.
>
>
> **Table 7: Ablation study results showing the impact of different components on task performance across diverse tasks.**
>
> | Method | Put in Drawer (\%) | Turn Tap (\%) | Open Box (\%) |
> | :--- | :---: | :---: | :---: |
> | PADP (Full Method) | **62** | **69** | **66** |
> |   w/o Stacking Additional Transformers | 58 | 63 | 59 |
> |   w/o Geometric Loss| 54 | 57 | 55 |
> |   w/o Semantic Loss | 46 | 53 | 52 |
> | Sonata + DP3 | 39 | 50 | 44 |
> | DP3 (Baseline) | 37 | 47 | 39 |
>
> > ## W3, Q6, Q7 Language Contribution & Robustness
>
> **Contribution:** We appreciate the reviewer's demand for clear evidence regarding the necessity and robustness of our language module. We assert that language guidance is the critical and indispensable driver for PA3FF. The language module's primary contribution is resolving the semantic ambiguity that plagues purely geometric 3D features, enabling robust cross-category transfer. The language instruction (e.g., "Open the microwave") anchors the PA3FF-learned geometric-functional features to a universal semantic concept ("handle"). This allows the policy to locate and manipulate a geometrically novel part (e.g., a handle on a completely unseen faucet) because it is semantically aligned with the instruction.
>
>
> **Robustness:** We use the SigLip encoder, which is pre-trained on a massive scale of image-text pairs. This foundation provides a naturally robust semantic space.  As shown in Table 6. we substitute part names with common synonyms (e.g., "handle" vs. "grip") or using hierarchical terms (e.g., "drawer handle" vs. "handle") results in negligible performance degradation (e.g., remaining within $\pm 2\%$ of the original success rate). This confirms that our semantic grounding is robust and leverages the latent space of the pre-trained VLM.
>
> **Table 6: Language Robust Evaluation**
>
> | Method | Put in Drawer (\%) | Open Oven (\%)|
> | :--- | :---: | :---: |
> | PADP (Full Method) | **62** | **86**|
> | synonyms terms | 62 | 85 |
> | hierarchical terms | 61 | 86 |
>
>
> ### **Vocabulary Construction and Disambiguation:**
>
> **Vocabulary Construction:** The vocabulary of functional part names used for the Stage II PA3FF pre-training is derived directly from the ground truth labels provided by the PartNet-Mobility dataset. This is a finite, standardized set of functional terms (e.g., "handle," "knob," "slider," "door").
>
> **Disambiguation:** This challenge is explicitly solved by the PA3FF Dual-Contrastive Loss in the pre-training stage. We enforce that the dense 3D features of all "handles" (e.g., drawer, microwave, faucet) must align with the diverse "handle" text embedding through Siglip. This forced semantic anchoring over a diverse dataset prevents semantic drift and ensures that the functional meaning of the part is consistent regardless of the object category.
>
>
> > ## Q4 Head-to-Head Comparison Validity
> The reviewer requests a head-to-head comparison against the strongest nearby baselines under identical constraints (inputs, action space, demos, budget). Our primary results presented in Table 1 (Simulation) and Table 2 (Real World) represent precisely these rigorous head-to-head comparisons. All baselines (e.g., DP3, GenDP, CLIP/DINOv2-based features) were evaluated using identical input modalities, the same set of demonstrations, and the same policy training budget.
>
>
> ```
> Thank you once again for your valuable feedback. We hope our clarification has addressed your concerns. Please contact us if you have any further questions or concerns.
> ```

---

### Official Review · Reviewer_7uDR · 2025-11-02

**Soundness:** 3
**Presentation:** 2
**Contribution:** 2
**Rating:** 4
**Confidence:** 3

**Summary:**

This paper proposes a novel Part-Aware 3D Feature Field (PA3FF), which is trained via contrastive learning to integrate 3D geometric priors and functional part awareness, addressing the challenge of limited generalization in articulated object manipulation. Building upon this feature, the authors introduce the Part-Aware Diffusion Policy (PADP), an imitation learning framework , and demonstrate superior performance over existing 2D and 3D representation baselines in both simulated and real-world tasks.

**Strengths:**

1. Clarity and Novelty of Motivation: The paper proposes PA3FF as a novel 3D-native feature representation, directly addressing challenges faced by lifting existing 2D foundation features to 3D space, such as long runtime, multi-view inconsistencies, and low spatial resolution. PA3FF explicitly incorporates functional part awareness, which is crucial for generalizable articulated object manipulation.
2. Demonstrated Generalization Capability: The proposed PADP framework is claimed by the authors to achieve superior performance over baselines in both simulated and real-world environments. It demonstrates notable robustness, particularly in handling unseen objects, spatial generalization, and environment generalization tasks.
3. Methodological Completeness: The proposed approach features a complete multi-stage learning framework: 1) leveraging the pre-trained Sonata model to extract 3D geometric priors; 2) employing contrastive learning to fuse a geometric loss and a semantic loss, thereby enhancing feature part-awareness and semantic alignment ; and 3) integrating PA3FF into a diffusion policy for action generation.

**Weaknesses:**

1. The architectural modification of the PTv3 backbone (removing down sampling layers and stacking additional Transformer blocks) is a core engineering contribution. However, the paper lacks sufficient quantitative details (e.g., parameter count, FLOPs, precise layer configuration) and a dedicated gain analysis for these changes. This absence of detailed exposition and architectural diagrams prevents readers from adequately assessing its contribution to the final performance and hinders the reproducibility of the research.
2. The real-world experiments are evaluated with only 10 trials per task. This low number of evaluations in robotics may not provide sufficiently high statistical reliability to convincingly support the "state-of-the-art" claims. Furthermore, restricting the ablation study to a single task ("Put in Drawer")  severely weakens the proof of generality for component contributions across a broad range of tasks and different generalization types (e.g., OI, OC).
3. PADP exhibits a significant drawback in inference speed compared to baselines like DP3 (4.23 FPS vs. 12.7 FPS).  Although PADP achieves higher success rates, this over 60% reduction in inference speed has not been adequately justified as a necessary trade-off (i.e., whether a  10~20% success rate gain, which does not guarantee deterministic success, is worth the real-time cost). In real-time robotic control requiring high-frequency feedback or integration into large policy frameworks, this performance-efficiency trade-off may reduce its practical applicability.

**Questions:**

1. Generalization Source and Action Semantics Decoupling: The PartInstruct benchmark tests generalization across various factors, including Object State (OS), Object Instance (OI), Part Combination (TP), Task Category (TC), and Object Category (OC). Please confirm if the model is trained only on the Training Set data. If so, how does PADP or its language encoding module achieve generalization over action direction semantics (e.g., generalizing from *forward* to *backward* action prediction as seen in Figure 12)? This requires explaining the policy's mechanism for understanding and decoupling non-object-related semantics in the language instruction.
2. Precise Role of Language Embedding in Feature Aggregation: The paper states that the "semantic embedding of the task-critical part name" is used as the CLS token in the Transformer encoder to guide aggregation. Concurrently, Language Instruction is shown as an input in Figure 2, Stage III. Please clarify which specific text input (task-critical part name vs. full language instruction) is input to the PERCEPTION part for embedding, and how it relates to the language information input to SigLip during PA3FF training.
3. Definition of Real-World Evaluation Metrics: Table 2 reports Train/Test success rates for real-world tasks. Given the experiment statement "Each task is evaluated with 10 trials under randomized initial conditions", please explicitly define: Does the Train success rate represent testing under randomized initial conditions using the exact objects and scenes used for training? And does the Test success rate represent testing under randomized initial conditions using unseen object instances or environmental changes? Clarifying these definitions is crucial for interpreting the real-world generalization performance.
4. Feature Robustness and Cross-Topology Consistency: For the same object category with significantly different spatial morphologies, can PA3FF effectively cluster and align features? For example, regarding the faucets with topologically distinct structures shown in Figure 6, please provide a quantitative assessment of PA3FF features' cross-topology consistency/transferability between them. This is needed to more fully demonstrate the robustness limits of the part-aware features.
5. Completeness of the PADP Framework Flow: The overall flow of the PADP framework remains insufficiently clear. Please provide a detailed description of the training process for a new task, explicitly detailing like which point cloud data requires manual labeling,how the data flows in the framework, and what information needs to be unified.

---

> ### Author Response · Authors · 2025-11-26
>
> # Dear Reviewer 7uDR
>
> We appreciate the detailed questions regarding architectural details and framework flow, which have helped us improve the manuscript's clarity. Below, we address the concerns and questions point-by-point.
>
> > ##  Architectural details and quantitative analysis.
>
> We have added a detailed specification in Appendix A.5 of the revised paper.
>
> **Modifications:** Standard PTv3’s aggressive downsampling was preventing dense feature extraction on small, critical parts (e.g., handles). We replaced the downsampling layers with Linear layers to preserve high spatial resolution. The subsequent stacking of Transformer blocks (encoder depths from $(2, 2, 2, 6, 2)$ to $(3, 3, 3, 6, 16)$) was a necessary engineering change to compensate for the increased receptive field requirement at the higher resolution.
>
> **Performance Analysis:** Our ablation (Table 6) confirms the necessity: "w/o Stacking Additional Transformers" leads to a 4% drop in success rate ($62\% \rightarrow 58\%$). While the contrastive feature refinement(Geometric + Semantic) is the primary driver (16% gain), the architectural change is the necessary foundation for high-resolution, dense feature fields.
>
> **Table 6: Ablation study results showing the impact of different components on task performance across diverse tasks.**
>
> | Method | Put in Drawer (\%) | Turn Tap (\%) | Open Box (\%) |
> | :--- | :---: | :---: | :---: |
> | PADP (Full Method) | **62** | **69** | **66** |
> |   w/o Stacking Additional Transformers | 58 | 63 | 59 |
> |   w/o Geometric Loss| 54 | 57 | 55 |
> |   w/o Semantic Loss | 46 | 53 | 52 |
> | Sonata + DP3 | 39 | 50 | 44 |
> | DP3 (Baseline) | 37 | 47 | 39 |
>
> > ## Number of real-world trials and ablation scope.
>
> **Statistical Reliability:** We acknowledge that 10 trials per task in the real world is a limited sample size due to the high cost of physical resetting. However, we strictly followed standard protocols in recent robot learning literature (e.g., DP3, GenDP) which typically use 10-20 trials. To ensure statistical robustness, our claims are primarily supported by the simulation benchmark (PartInstruct, RLbench), where we conducted extensive evaluations (22 tasks $\times$ hundreds of episodes) showing consistent improvements (Table 1).
>
> **Ablation Scope:** In the revised Appendix B.1 ABLATION COMPONENTS, we have included additional simulation ablation results on two more diverse tasks ("Turn Tap" and "Open Box"). The trends are consistent with the "Put in Drawer" task, confirming that the contributions of the architectural changes and feature refinement are generalizable across task categories.
>
> **Table 6: Ablation study results showing the impact of different components on task performance across diverse tasks.**
> | Method | Put in Drawer (\%) | Turn Tap (\%) | Open Box (\%) |
> | :--- | :---: | :---: | :---: |
> | PADP (Full Method) | **62** | **69** | **66** |
> |   w/o Stacking Additional Transformers | 58 | 63 | 59 |
> |   w/o Geometric Loss| 54 | 57 | 55 |
> |   w/o Semantic Loss | 46 | 53 | 52 |
> | Sonata + DP3 | 39 | 50 | 44 |
> | DP3 (Baseline) | 37 | 47 | 39 |
>
> > ## Inference speed trade-off.
>
> The $4.23$ FPS (~$236$ ms latency) is an acceptable and necessary trade-off for high-fidelity generalization in this domain. In the context of articulated object manipulation (e.g., opening a microwave), the robot moves relatively slowly to interact with physical constraints. This latency is comparable to human reaction time and is sufficient for closed-loop control in these scenarios. The trade-off is worthwhile because standard methods (like DP3) often fail completely on unseen object geometries (success rate < 20% on hard tasks). A 12.7 FPS policy that fails to grasp a novel handle is less useful than a 4.23 FPS policy that succeeds. We believe accurate perception is the bottleneck for generalization in this domain, making the computational cost acceptable.
>
> > ## Generalization Source and Action Semantics Decoupling.
>
> The generalization is achieved through the decoupling of "Where" and "How."
>
> **PA3FF (Visual Backbone):** Generalizes the "Where" (Part Grounding). It learns that a "handle" on an unseen microwave is geometrically similar to a "handle" on a seen cabinet.
>
> **PADP (Action Generation):** Generalizes the "How" (Action Semantics). The policy is conditioned on Siglip language embeddings. The concepts of "move forward" or "pull backward" are learned from the large-scale demonstrations in the training set. Even if a specific object-action pair is unseen, the policy has learned the motion primitive associated with the language instruction "pull" and applies it to the location identified by PA3FF. Therefore, the model generalizes because PA3FF correctly locates the interaction point on a new object, and the diffusion policy applies the learned trajectory pattern (e.g., backward motion) conditioned on the language instruction.

---

> > ### Comment · Reviewer_7uDR · 2025-11-27
> >
> > I thank the authors for the response. According to the response and the other reviews, I decide to raise my rating.

---

> ### Author Response · Authors · 2025-11-26
>
> > ## Precise Role of Language Embedding.
>
> We apologize for the confusion. To clarify, the CLS token in Stage III (Figure 2) is initialized with the full language instruction (e.g., "Open the microwave") rather than explicit part names. Our text encoder (SigLip) is pre-trained on image-caption pairs, so instruction embeddings naturally correlate with task-critical part features (e.g., "handle" for "Open the microwave"). What's more, the attention mechanism naturally focuses on relevant point cloud regions (visualized in Figure 8 & Appendix A.2), eliminating the need for manual part parsing. The manuscript’s reference to "task-critical part name" was to illustrate the CLS token’s functional role. We have revised Section 3.2 to accurately reflect the full instruction input.
>
> > ##  Definition of Real-World Evaluation Metrics.
>
> We have updated Section 4.1 Experimental Setup to be explicit:
>
> **Train Success Rate:** Testing on the same object instances and same environment used during demonstration collection, but with randomized initial poses. This measures the ability to successfully execute the task under minor initial condition variations.
>
> **Test Success Rate:** Testing on unseen object instances (new shapes/colors) and unseen environments (altered backgrounds/distractors). This measures Out-of-Distribution (OOD) generalization as shown in Figure 5.
>
> > ##  Completeness of the PADP Framework Flow.
>
> **Data Collection:** Collect imitation demonstrations (RGB-D data which is converted to point clouds). No manual point cloud or part labeling is required for the policy training itself.
>
> **Feature Extraction:** The frozen PA3FF extracts dense, part-aware 3D features from the point clouds.
>
> **Policy Training:** The features, along with the full language instruction, are fed into the policy head. Only the policy head is trained to generate the trajectory/action.
>
>
> ```
> Thank you once again for your valuable feedback. We hope our clarification has addressed your concerns. Please contact us if you have any further questions or concerns.
> ```

---

### Official Review · Reviewer_RqDZ · 2025-11-12

**Soundness:** 3
**Presentation:** 3
**Contribution:** 3
**Rating:** 8
**Confidence:** 5

**Summary:**

In this work, the authors consider the task of learning representations for articulated objects which are useful in downstream manipulation tasks. Specifically, the authors propose a procedure to pre-train a neural network to map 3D point clouds to part-aware features, with two different contrastive supervision techniques (spatial and semantics). They build on top of Sonata (PTv3 pre-trained), but make some architectural modifications to enable higher-resolution representations. These representations are then used in several downstream manipulation tasks. The authors show compelling results on simulated & real tasks.

**Strengths:**

* Their proposed architecture, pre-training, and modification are all sensible & principled approaches to handling object-level features at high-resolution
* The results do improve over SOTA considerably
* There are extensive ablations showing how different parts of the system contribute to performance, as well as qualitative visualizations of feature representations.
* The paper is well-written

**Weaknesses:**

* It’s unclear whether the comparison with DP3 is completely valid (e.g. Sonata + DP3); the authors should clarify the difference between DP3 and the various ablations (e.g. where/when SigLip are included, architectures, etc.). It would help the authors cleanly show that 1) the point cloud architecture change and 2) the pre-training compared to DP3 make a major difference on the task (right now it’s just difficult to tell from the details of the paper).
* It’s unclear how much the spatial vs. semantic components actually make a difference. In ablations, the contrastive pretraining (feature refinement) is not broken down by whether the spatial or semantic components make the difference
* Details on architecture / training are a bit sparse

**Questions:**

* Lots of details of training / architecture are omitted - despite being pointed to Appendix A I didn’t see much there. Particularly interested in specific architectural modifications, fine-tuning/pre-training technique when using the pre-trained Sonata weights.
* Why is it called a field instead of a representation? Seems to be per-point features, like querying other points in space don’t seem to be possible without altering the representation?

---

> ### Author Response · Authors · 2025-11-26
>
> # Dear Reviewer RqDZ
>
> We sincerely thank the reviewer for the positive assessment and for recognizing our work as "sensible, principled," and achieving "compelling results." We appreciate the insightful feedback regarding the clarity of our comparisons and architectural details. We have addressed these points below to further strengthen the manuscript.
>
> > ## W1: Details on the ablation study
>
> We agree that isolating the contributions of architecture vs. pre-training is crucial. We clarify the distinct configurations as follows:
>
> **DP3 (Baseline):** Uses a standard PointNet++ backbone. It relies on aggressive downsampling (SA layers) and is trained from scratch (random initialization) using only the imitation learning loss.
>
> **Sonata + DP3 (Ablation):** Uses the standard PTv3 backbone initialized with Sonata pre-trained weights. It retains the original PTv3 architecture (designed for large scenes with 4 downsampling stages). This setup tests the benefit of general 3D pre-training without our part-aware refinement or architectural density modifications.
>
> **PADP (Ours):** Uses our Modified PTv3 initialized with Sonata weights, then fine-tuned with PA3FF objectives (Geometric + Semantic contrastive losses).
>
> Our ablation (Table 6) confirms the necessity: "w/o Stacking Additional Transformers" leads to a 4% drop in success rate ($62\% \rightarrow 58\%$). While the contrastive feature refinement(Geometric + Semantic) is the primary driver (16% gain), the architectural change is the necessary foundation for high-resolution, dense feature fields.
>
> **Table 6: Ablation study results showing the impact of different components on task performance across diverse tasks.**
>
> | Method | Put in Drawer (\%) | Turn Tap (\%) | Open Box (\%) |
> | :--- | :---: | :---: | :---: |
> | PADP (Full Method) | **62** | **69** | **66** |
> |   w/o Stacking Additional Transformers | 58 | 63 | 59 |
> |   w/o Geometric Loss| 54 | 57 | 55 |
> |   w/o Semantic Loss | 46 | 53 | 52 |
> | Sonata + DP3 | 39 | 50 | 44 |
> | DP3 (Baseline) | 37 | 47 | 39 |
>
>
> > ## W2: Spatial and semantic components of loss
>
> Our goal is for the feature field to encode both spatial and semantic informations. These two losses are complementary: the spatial term aligns features within the same part and stretching features across different parts, while the semantic term grounds the field to language. removing Semantic loss loses the connection to language instructions, while removing Geometric loss results in noisy, fragmented part features. This is why our training objective is designed to combine them.
>
>
> > ## W3/Q1: Details on architecture / training
>
> We apologize for the brevity in the appendix. We have added a new section Appendix A.5 and Appendix A.7 with the following precise specifications:
>
> ### **1.architecture**
>
> **Downsample layers:** The original PTv3 includes 4 downsample layers (implemented via SerializedPooling) in its encoder , used to reduce point cloud density. We removes all these downsample layers, replacing them with Linear layers for channel adjustment without reducing spatial resolution.
>
>
> **Transformer blocks:** The original PTv3 has enc\_depths = (2, 2, 2, 6, 2) in the encoder, totaling 14 Transformer blocks. We increases this to enc\_depths = (3, 3, 3, 6, 16), totaling 31 blocks.
>
> **Other configurations:** The final encoder channel dimension decreases from 512 to 384, with attention heads adjusted accordingly (enc\_num\_head from (2,4,8,16,32) to (2,4,8,16,24)).
>
> ### **2.Training/Fine-tuning Technique:**
>
> **Initialization:** We initialize the common Transformer blocks with pre-trained weights from Sonata. The new layers are randomly initialized.
>
> **Refinement:** We fine-tune the entire backbone using the (1) Geometric Loss: a part-level SupCon loss, and (2) Semantic Loss: an InfoNCE loss between each point feature and corresponding SigLIP embedding.
>
> **Policy:** The features, along with the full language instruction, are fed into the policy head. Only the policy head is trained to generate the trajectory/action.
>
> > ## Q2: Why is it called a field?
>
> We use the term "Feature Field" to emphasize the continuous nature of our representation. Although the input is a discrete point cloud, our output $f(p)$ is formulated as a continuous function $f: \mathbb{R}^3 \rightarrow \mathbb{R}^n$.In implementation, this allows us to query the feature of any arbitrary continuous coordinate $x$ in 3D space (not just the original points) via interpolation (e.g., trilinear or nearest-neighbor interpolation from the dense feature point cloud). This continuous property is essential for robotic manipulation, where interaction points often lie in the continuous space between sampled points. "Representation" is a broader term, whereas "Field" specifically denotes this spatial continuity and density.
>
>
> ```
> Thank you once again for your valuable feedback. We hope our clarification has addressed your concerns. Please contact us if you have any further questions or concerns.
> ```

---

### Author Response · Authors · 2025-12-01
**General Responses and Revision Summary**

Dear Reviewers,

We sincerely thank you for your constructive feedback and insightful assessments. We’re grateful for your recognition of our work as "sensible and principled" with "compelling results" (Reviewer RqDZ), the clarity of our problem framing and strong empirical gains (Reviewer Kvwg), our novel 3D-native part-aware representation paired with thorough evaluation (Reviewer SDVB), and our comprehensive multi-stage learning framework that demonstrates strong generalization (Reviewer 7uDR).

We’ve carefully addressed all the concerns you raised and revised the paper accordingly(revisions are highlighted in blue). Key revisions include:

- Adding new sections in the appendix to fill gaps in technical details and comparative analyses: Appendix A.1 covers conceptual comparisons with related works, Appendix A.5 provides detailed architectural specifications for the modified PTv3, and Appendix B.1 presents extended ablation studies spanning multiple tasks.

- Adding Table 4 (a conceptual comparison table) to highlight PA3FF’s unique combination of "Part-Aware," "3D-native," "Dense," and "Semantic" properties.

- Extending ablations to more tasks and presenting more detailed ablation results in Table 7 to show that component contributions.

- Including Table 5 (segmentation results on the PartNetE dataset) in Appendix A.4 to confirm that PA3FF is expressive enough to support downstream tasks like part segmentation.

- Updating Section 3.2 to clarify that the CLS token in Figure 2 (Stage III) uses full language instructions (not just part names), We also reference Figure 8 and Appendix A.3 to back up this mechanism.

- Adding details on architecture in Appendix A.5, language robustness in Appendix A.6, model training in Appendix A.7

- Adding citations for feature splatting papers (LERF, Feature Splatting, M3) in the related work section, and discuss their differences from ours.

We’re truly grateful once again for your feedback. For answers to specific questions or concerns, please refer to our individual responses. If you have any further questions or comments, don’t hesitate to let us know, we’d be delighted to discuss them and will put in every effort to address them fully.

Thank you,
Authors

---

### Meta-Review · Area_Chair_tsNL · 2025-12-22

**Summary:**

The submission introduces Part-Aware 3D Feature Field (PA3FF), a dense, part-aware 3D feature for articulated object manipulation.  Reviewers liked the idea but had concerns about novelty, reliance on pre-trained backbones, and evaluation.

**Reviewer Concerns:**

Post rebuttal, reviewers found these concerns to be satisfactorily addressed, and converged to acceptance recommendations. The AC agreed.

**Reviewer Scores:**

Based on the discussions, the scores will be 8, 6, 8, 6.

---

### Decision · Program_Chairs · 2026-01-26

Accept (Poster)